# RoleArena: A Multi-Agent Role-Playing Environment for Long Multi-Turn Dialogues with Autonomous Plot Progression

## Abstract

Improving the plot progression ability of role-playing language agents (RPLAs) plays a critical role in enhancing user experience and dialogue immersion. However, research on plot progression in role-playing language agents is still limited. Existing datasets often contain too few dialogue turns and lack large-scale sentences with clear deterministic plot turns, which restricts further model development and evaluation. To bridge this gap, we propose *RoleArena*, an innovative multi-agent role-playing environment. By introducing a critic agent and combining discrete plot lines with environment-agent-based dynamic judgments, the system generates long multi-turn dialogues with key markers for plot progression. Based on *RoleArena*, we construct the PlotStream dataset, which contains 39.3k characters and 170k dialogues, with annotations for key plot progression sentences. In addition, based on PlotStream, we further develop PlotRole-7B and PlotRole-72B, RPLAs built upon the Qwen2.5 model and specialized in plot progression capabilities. Extensive experiments show that *RoleArena* and PlotRole improve dialogue generation with plot progression and strengthen the ability of plot progression. The code and data is available at https://anonymous.4open.science/r/RoleArena-CB22.

## 1 Introduction

The rapid development of large language models (LLMs) accelerates the digital transformation of human society (Piatti et al., 2024; Zhou et al., 2025). As research on LLMs deepens, there is increasing interest in the human-like intelligent traits they exhibit (Thoppilan et al., 2022). As a result, role-playing language agents (RPLAs) have gained attention and become a key focus of current research (Chen et al., 2024; Shao et al., 2023). RPLAs introduce innovative interaction experiences between humans and artificial intelligence (Santurkar et al., 2023; Zhao et al., 2024), leading to numerous application scenarios, such as role-playing game dialogues and digital human interactions (Wang et al., 2025b; Li et al., 2023; Ashby et al., 2023; Yao et al., 2024). In such dialogue environments, users can engage in immersive dialogue scenarios (Tao et al., 2024a). Therefore, RPLAs face the challenge of providing users with an immersive chat experience, necessitating the enhancement of their plot progression capabilities (Wakaki et al., 2024).

However, current research on improving the plot progression capabilities of RPLAs is insufficient. Specifically, there are certain limitations: 1. Insufficient dialogue depth: Existing RPLAs typically have short multi-turn dialogue sequences, often limited to shallow interactions within a dozen turns, which fail to reflect RPLAs' performance in complex multi-turn dialogue scenarios. 2. Lack of plot progression: Datasets lack long dialogues with key plot progression, making it difficult for RPLAs to learn how to proactively and logically advance the plot, leading to monotonous and repetitive dialogues. Although some work attempts to provide datasets with plot progression statements (Tu et al., 2024; Lu et al., 2025; Yuan et al., 2024), these require significant human effort to identify key plot points, resulting in high costs and making large-scale generation impractical (Zhang et al., 2025).

In this paper, we introduce RoleArena, a multi-agent role-playing environment capable of generating long, multi-turn dialogues with autonomously developed plotlines. RoleArena allows multiple

RPLAs to interact in fictional dialogue scenarios. Specifically, it contains three types of agents: **Character agent**, which has independent memory and engage in dialogue with other role agents according to the environment agent's guidance. **Environment agent**. To ensure a coherent theme and drive the plot forward, environment agents first generate the environment, dialogue themes, and a discretized plotline. After each interaction by the role agents, the environment agent determines whether to advance the plot. The environment agent focuses on the current plot point. If it chooses to advance the plot, it generates a naturally coherent plot progression based on the preceding and following events. This approach significantly improves the robustness of plot-driving statements compared to prior work. **Critic agent.** To achieve controlled multi-turn dialogues, we introduce a critic model to perform a secondary evaluation of the environment agent's decisions. To improve system efficiency, the critic agent only initiates the secondary evaluation when the environment agent chooses to advance the plot. The critic agent adjusts the environment agent's decision based on the current and required total number of dialogue turns. The plot only advances if the critic agent agrees with the environment agent's decision.

The main difference between RoleArena and existing work is that: 1) RoleArena uses discretized plotlines to drive the environment agent to dynamically determine plot advancement points during multi-turn dialogues and generate natural plot progressions, offering a robust method for plot-driving data generation. 2) RoleArena introduces critic agents to make the overall dialogue turn count controllable, while ensuring sufficient development of each plot point.

Based on RoleArena, we construct a large-scale multi-turn dialogue dataset, PlotStream, which includes plot-driven annotations. To investigative the enhancement of plot-driving capabilities in RPLAs, we develop PlotRole-7B and PlotRole-72B based on PlotStream. Additionally, we propose the RoleArena-Eval evaluation system, based on five dimensions, to comprehensively assess RPLAs' abilities in long-turn dynamic dialogue environments.

Briefly, our main contributions are three-fold:

- We propose the multi-agent environment RoleArena, designed to generate controllable-turn, plot-driven RPLAs multi-turn dialogues, and build PlotStream, a large-scale dataset with plot-driving annotations for developing and evaluating RPLAs.

- We develop PlotRole-7B and PlotRole-72B to better enhance the plot progression capability of RPLAs.

- We propose RoleArena-Eval, which comprehensively evaluates RPLAs' abilities in complex dynamic multi-turn dialogues over five dimensions. In particular, RoleArena-Eval introduces a quantitative assessment of plot progression.

## 2 RELATED WORK

**Multi-agent role-playing language agent dialogues.** Several agents are assigned to play different roles and are used to simulate plotlines and generate dialogues. CharacterBox (Wang et al., 2024a) proposed a simulation sandbox that introduces a narrator agent and character agents, where conversations between character agents and their interactive feedback with the narrator drive the dialogue forward. COSER (Wang et al., 2025b) built a multi-agent environment for dialogue simulation, in which an environment model is introduced to provide feedback. IBSEN (Han et al., 2024) attempted to base its design on a fictional story outline, using agents to play the roles of director and actors to perform the plotline. BookWorld (Ran et al., 2025), grounded in drama theory, uses multiple agents across different scenes to complete a story. These methods confirm the effectiveness of multi-agent collaboration in role-playing agent dialogues; however, research on narrative progression is still insufficient. RoleArena introduced a critic agent, achieving more stable and controllable dialogue turns while emphasizing narrative progression.

**Construction of role-playing datasets.** Current methods for building role-playing datasets mainly follow two approaches: obtaining character configuration metadata, dialogues, and other contextual information from real texts such as novels and film scripts, which allows role-playing language agents to act as the characters contained within them and provide users with an immersive conversational experience (Li et al., 2023; Dai et al., 2024; Shao et al., 2023; Zhou et al., 2024). Creating synthetic character configuration metadata and generating simulated dialogues based on it greatly

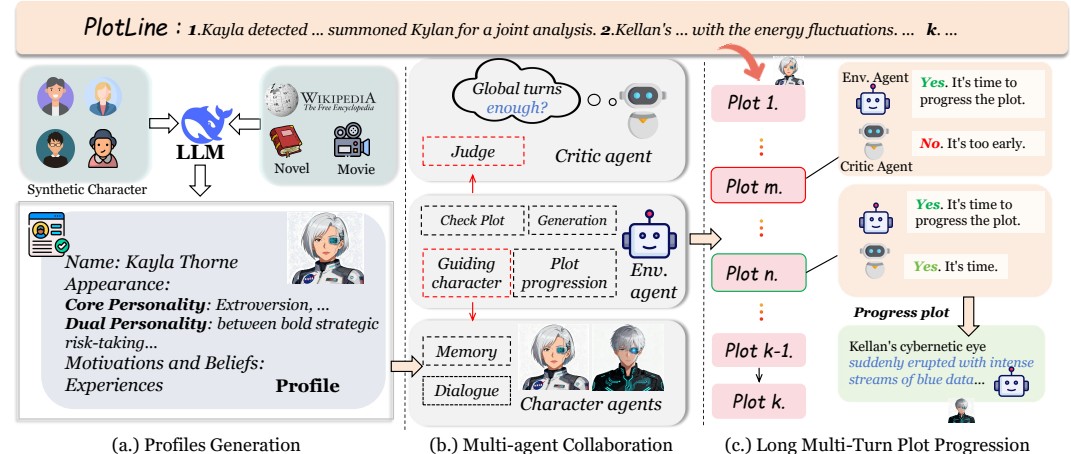

Figure 1: Overview of the RoleArena multi-agent environment. After generating character profiles in (**a.**), we drive the interactions of different agents in (**b.**), where the Env. agent sets up the plot and guides the Character agents to perform multi-turn dialogues in (**c.**) to achieve plot progression, while the Critic agent oversees the overall dialogue rounds, ensuring controllable dialogue turns and improving system robustness.

expands the sources of data for constructing role-playing datasets, significantly enhancing the role generalization ability of role-playing language agents while avoiding infringement risks associated with some literary and film works (Ge et al., 2024; Lu et al., 2025; Wang et al., 2025a; Tao et al., 2024b). PlotStream is mainly built on synthetic characters.

**Role-playing evaluation.** Evaluating the performance of role-playing is a challenging task, as a comprehensive evaluation often requires considering various subjective indicators. InCharacter (Wang et al., 2024b) proposes using personality tests to assess the fidelity of RPLA. DITTO (Lu et al., 2024) evaluates role-playing language agents by judging character consistency, knowledge accuracy, and refusal rates. In addition to evaluating static character fidelity, consistency, and personification, CharacterBox (Wang et al., 2024a) extends the evaluation to specific actions in dynamic scenes. COSER (Wang et al., 2025b) proposes a GCA-based evaluation method, where LLMs perform role-playing and multi-round dialogues in a given scene and are evaluated across four dimensions: story quality, personification, character fidelity, and story consistency. CharacterEval (Tu et al., 2024) designed a four-dimensional evaluation system, including dialogue ability, character consistency, role-playing appeal, and personality backtesting. Beyond the comprehensive evaluation of character actions, psychology, and dialogue, RoleArena-Eval also extends the evaluation to plot progression and evolution.

## 3 ROLEARENA

### 3.1 PRELIMINARY

**Motivation.** The ability of role-playing language agents to drive the plotline greatly affects the user's immersive experience, but research in this area has not made significant progress so far. Role-PlotZhang et al. (2025) proposes a method based on triggering the plotline transition subspace, which is effective, but the data used for training often requires manual filtering, resulting in high labor costs, and it heavily depends on historical dialogue information of the characters. In addition, the dialogue turns and plotline development in existing datasets are relatively limited, making it difficult to support complex multi-turn dialogues. Research on generating robust long-turn multi-round dialogues will fill the gap in this area of study.

**Role-playing multi-round dialogue.** In a given environment, if there are $M$ dialogue agents, we define the completion of the dialogue by all agents as the end of one dialogue turn.

**Plot progression.** The process unfolds gradually with changes in character actions and event developments, with each step laying the foundation for the subsequent plot, maintaining the continuity and appeal of the story, and driving it in new directions.

## 3.2 AGENT ARCHITECTURE

RoleArena is a framework for multi-agent collaboration that drives natural story progression and maintains long-turn conversations. It mainly includes three functional agents: environment agent $\mathbb{E}$, character agent $\mathbb{R}$, and critic agent $\mathbb{C}$.

### 3.2.1 ENVIRONMENT AGENT

The environment agent, as a core functional agent in RoleArena, is mainly responsible for four sub-tasks:

**1) Topic, environment, and plotline generation**. Before the start of the entire conversation $D$, the environment agent $\mathbb{E}$ first generates the story topic, environment, and corresponding plotline based on the character agents $[R_1, R_2, ..., R_M]$ present in the current scene. The plotline is then discretized into a sequence of nodes $< S_1, S_2, ..., S_k >$. To achieve controllable turns, we introduce a hyperparameter $r$, which defines the minimum number of conversation turns required for the story. $\mathbb{E}$ will consider $r$ when generating the plotline length $k$.

**2) Guiding character agents**. When the conversation reaches the $i$-th round, after each action $r$ of a character agent $\mathbb{R}$ is completed, $\mathbb{E}$ immediately receives the action information of that character and prioritizes prompting the silent character agent $\mathbb{R}$ to take action. $\mathbb{E}$ will add appropriate narration content $n$ and guide the actions of the character agent. Therefore, the overall sequence of the $i$-th round of conversation is

$$[n_1, r_1, n_2, r_2, \ldots, n_l] \quad \text{or} \quad [r_1, n_1, r_2, n_2, \ldots, n_l] \tag{1}$$

The reason for the two different situations is that at the start of the first round of conversation, the environment agent $\mathbb{E}$ initiates, and at the end of each round, the environment agent provides immediate feedback on the actions of the character agent $\mathbb{R}$.

**3) Check the story's end state and drive progression**. In complex story tasks, purely relying on character dialogues to advance the plot often leads to stagnation or deviation from the main plotline. Unlike previous methods, in RoleArena, the environment agent $\mathbb{E}$ dynamically determines the story's end state in real-time. After each action by a character agent R, the environment agent E combines the character's dialogue history $\mathcal{H}$, the current story node's dialogue turn $d_k$, the current story node $S_u$, and the next story node $S_{u+1}$ to decide whether to end the current plotline $S_u$ and move the plot to $S_{u+1}$.

**4) Robust plot progression**. Once the environment agent $\mathbb{E}$ and the critic agent $\mathbb{C}$ reach a consensus among the characters and agree to advance the plot, $\mathbb{E}$ will be instructed to create a natural plot twist narration $n_i^*$ for the plotline $< S_u, S_{u+1} >$. This plot twist narration $n_i^*$ will also direct the next character agent to generate the corresponding action $r_i^*$.

In our design, plot twists are actively judged and triggered by the environment agent. When the system detects the need to advance the plot, the environment agent generates a plot twist narration based on the established story nodes and task requirements, guiding the characters into a new story phase. Compared to methods relying on manual filtering, the advantages of this mechanism include: *1)* Robustness: plot twists are based on clear task signals and node logic, avoiding random or inconsistent progression, ensuring the controllability and coherence of story development. *2)* Cost efficiency: Through automated twist generation, the system can significantly reduce manual intervention and data filtering costs, enabling larger-scale and more reusable plot generation processes.

### 3.2.2 CHARACTER AGENT

To make the overall conversation controllable and improve its quality, each action of the character agent $\mathbb{R}$ is guided by the environment agent $\mathbb{E}$. At the same time, $\mathbb{R}$ needs to consider its dialogue history with other character agents when taking actions. Furthermore, to more realistically reflect the character's inner thoughts and enrich the emotional expression of the character agent, we require $\mathbb{R}$ to express $< thoughts >$, $(actions)$, and speech while taking actions.

Table 1: Comparison between the PlotStream dataset and other existing datasets. Num., #Turns, #Conv., and Multi-Chara. denote the number of characters, the number of dialogue turns, the total scale of dialogues, and whether multiple characters are supported, respectively.

| Dataset | Character | | Conversation | | | Dialogue | | | Plot |
|---|---|---|---|---|---|---|---|---|---|
| | Num. | Profile | #Turns | #Conv. | Multi-Chara. | Speech | Thought | Action | Progre. |
| Charater-LLM Shao et al. (2023) | 9 | ✓ | 13.2 | 14,300 | | ✓ | | | |
| ChatHaruhi Li et al. (2023) | 32 | ✓ | >2 | 54,726 | ✓ | ✓ | | | |
| RoleLLM Wang et al. (2023) | 100 | ✓ | 2 | 140,726 | | ✓ | | | |
| CroSS-MR Yuan et al. (2024) | 126 | ✓ | 2 | 445 | | | | | |
| CharacterGLM Zhou et al. (2024) | 250 | ✓ | 15.8 | 1,034 | | ✓ | | | |
| CharacterEval Tu et al. (2024) | 77 | ✓ | 9.3 | 1,785 | | ✓ | | ✓ | |
| DITTO Lu et al. (2024) | 4,002 | ✓ | 5.1 | 7,186 | | ✓ | | | |
| MMRole Dai et al. (2024) | 85 | ✓ | 4.2 | 14,346 | | ✓ | | | |
| CharacterBench Zhou et al. (2025) | 3,956 | ✓ | 11.3 | 13,162 | | ✓ | | | |
| COSER Wang et al. (2025b) | 17,966 | ✓ | 13.2 | 29,798 | ✓ | ✓ | ✓ | ✓ | |
| RolePlot Zhang et al. (2025) | 20 | ✓ | >2 | 28,127 | | ✓ | | | ✓ |
| RoleMRC Lu et al. (2025) | 10,200 | ✓ | 9.5 | 37,900 | | ✓ | | | |
| PlotStream | **39,312** | ✓ | **>50** | 170,148 | ✓ | ✓ | ✓ | ✓ | ✓ |

### 3.2.3 CRITIC AGENT

Relying solely on the environment agent to judge plot progression can lead to the plot advancing too quickly, which may affect the quality and coherence of the conversation. To improve the system's robustness, we introduced the critic agent $\mathbb{C}$ to assist the decision-making process of the environment agent. Specifically, when the environment agent determines that the plot should advance, $\mathbb{C}$ evaluate this decision. At this point, the critic agent not only considers the current plot state but also takes into account the remaining dialogue turns $(r - i)$ and the remaining story elements $(k - u)$ to ensure the plot's progression is reasonable. The system will only update the current plot state $S_u$ to $S_{u+1}$ when both the environment agent $\mathbb{E}$ and the critic agent $\mathbb{C}$ agree that the plot can advance. If the critic agent believes that the plot is advancing too quickly and the current plot is not sufficiently developed, it will reject the environment agent's judgment, and the system will maintain the current plot state until the conditions for progression are met.

$$S_u \xrightarrow{\mathbb{E} \wedge \mathbb{C}} S_{u+1} \quad \text{if} \quad \mathbb{P}(\mathbb{E}, \mathbb{C}) = 1 \tag{2}$$

where $\mathbb{P}(\mathbb{E}, \mathbb{C})$ denotes the probability that both the environment agent $\mathbb{E}$ and the critic agent $\mathbb{C}$ agree on the plot progression. This design enhances the system's robustness and conversation quality by introducing the critic agent. The critic agent $\mathbb{C}$ assists the environment agent's judgment by evaluating the remaining turns and plot complexity, preventing the plot from advancing too quickly, thus improving dialogue stability. The dual evaluation mechanism ensures the reasonable timing of plot twists, avoiding fragmentation or rushed progression. Additionally, $\mathbb{C}$ considers both the remaining dialogue turns and plot development, ensuring that the dialogue progress matches the overall story pace, preventing the plot from advancing too early or too late.

## 4 DATASET AND AGENT TRAINING

### 4.1 PLOTSTREAM DATASET

Based on RoleArena, we build the first long-turn role-playing multi-turn dialogue dataset with plot progression, PlotStream. This section introduces the details of PlotStream.

### 4.1.1 DATASET CHARACTERISTICS

Compared to previous work, the PlotStream dataset has several features: *1)* **Long-turn dialogue rounds**. The average number of dialogue rounds in PlotStream exceeds 50, covering more complex and continuous multi-turn interactive scenarios, allowing for better simulation of real-world narrative and character interactions. *2)* **Clear annotation of key plot-driving statements**. Statements that drive plot twists and progress in the dialogue are clearly marked, making PlotStream an effective resource for improving the plot-driving ability of role-playing agents. *3)* **Rich language and character information**. PlotStream supports both Chinese and English dialogue data. To provide

more diverse data, PlotStream includes 10 styles of synthetic characters and also introduces novel characters based on human-created literature.

### 4.1.2 ROLE PROFILE CONSTRUCTION

We use Deepseek V3.1[1] to generate meta-configuration information for characters based on the large-scale "one-sentence synthetic characters" provided in PersonHub[2] (Ge et al., 2024). To increase the diversity of the data, we guide the LLM to generate 10 types of characters with highly distinguishable styles. Additionally, we extract 77 novel characters based on human-created literature and their corresponding information from CharacterEval[3] and COSER[4].

### 4.1.3 ROLEARENA DIALOGUE SIMULATION

Using the RoleArena multi-agent role-playing dialogue environment we proposed, based on the profiles of synthetic and novel characters, we use DeepSeek V3.1 to simulate long-turn role-playing multi-turn dialogues with plot-driving statement markers.

In this process, we perform a secondary check on PlotStream, filtering out samples with generation failures and overly long generated texts, ensuring the quality of the dataset.

### 4.2 PLOTROLE TRAINING

We use the constructed PlotStream dataset to train the LLM to improve its plot progression ability in role-playing. To improve the performance of role-playing agents in both narration generation and character response generation, we define two optimization goals for the role-playing agent's parameter training: 1) Narration target $n^*$, with its context during training as $[i_n, n_1, r_1, n_2, r_2, ..., r_i]$, where $i_n$ is the role-playing instruction for the task, including detailed information about the characters in the scene. $n^*$ is considered as the optimized plot-driving output. 2) Character response target $r^*$, with its context during training as $[i_r, ..., n_i, r_i, n^*]$, where $i_r$ is the role-playing instruction that includes the agent's role and other characters' information. $r^*$ is considered as the optimized character plot-driving response. The character $r^*$ response target represents the specific reactions of characters during critical plot advancements, including thoughts, actions, and speech.

$$\theta^* = \arg\min_{\theta} \left[ \underbrace{\mathbb{E}_{C_{n^*}} \, \mathcal{L}_{\text{CE}}\big(f(C_{n^*}; \theta), n^*\big)}_{\text{Narration Guidance}} + \underbrace{\mathbb{E}_{C_{r^*}} \, \mathcal{L}_{\text{CE}}\big(f(C_{r^*}; \theta), r^*\big)}_{\text{Role Response}} \right], \qquad (3)$$

where $C_{n^*} = [n, r, \ldots, n, r]$ denotes the dialogue context before narration, and $C_{r^*} = [n, r, \ldots, n, r, n^*]$ denotes the context including the narration.

## 5 ROLEARENA-EVAL

To fully assess the role-playing ability of role-playing agents in long-turn dynamic dialogue environments, inspired by previous work, we design the evaluation system RoleArena-Eval, which includes five dimensions.

1) **Dialogue Fluency (DF)**. This evaluates the smoothness and quality of the dialogue. The criteria include: Naturalness. Whether the dialogue is smooth and free from awkward or repetitive expressions. Coherence. Whether each round logically follows the previous one, without irrelevant responses or missing key information. Consistency. Whether a character's statements are consistent within the same or adjacent turns.

2) **Character Fidelity (CF)**. This evaluates how well the character's words and actions match their background, personality, and abilities. The criteria include: Knowledge. Whether the character's

---

[1] https://huggingface.co/deepseek-ai/DeepSeek-V3.1

[2] https://github.com/tencent-ailab/persona-hub

[3] https://github.com/morecry/CharacterEval

[4] https://github.com/Neph0s/CoSER

speech matches their identity, history, and known information. Personality. Whether the character's language style, decisions, and actions fit their core traits. Behavior. Whether the character's habits and actions are consistent.

3) **Emotional Expression (EE)**. This evaluates how deeply and authentically the character expresses emotions and interacts with others' emotions. Specific criteria include: Personality: Whether the character stays true to their personality traits in all conversations and actions. Empathy: Whether the character understands and appropriately responds to others' emotional states.

4) **Story Quality (SQ)**. This evaluates the quality of the story formed through the dialogue. Criteria include: Suspense: Whether the dialogue creates suspense, conflict, or tension to keep the user engaged. Theme: Whether the dialogue centers around a clear theme or goal.

5) **Plot Progression (PP)**. This evaluates how the plot progresses and whether it develops naturally. Criteria include: Natural Progression: Whether the plot changes are logically driven by character interactions and events, not forced. Proactivity: Whether the character makes active decisions, suggests plans, or takes actions to change the situation. Contribution: Whether each dialogue meaningfully contributes to the plot, avoiding repetition.

We design a quantitative evaluation method based on the Likert scale (Likert, 1932) for this evaluation system. For each specific evaluation criterion, the score range is set as $[1, 2, 3, 4, 5]$. Additionally, following a critique-based expert judgment approach, we first use GPT-4o[5] to generate critique opinions for the different dimensions, which are then taken into consideration during the final evaluation. RoleArena-Eval provides a comprehensive and complex evaluation system for role-playing, especially introducing an evaluation of plot progression, which is particularly important for enhancing the immersive experience of role-playing.

## 6 EXPERIMENTS

### 6.1 EXPERIMENTAL SETTINGS

**Test Set Selection and Evaluation Rounds Setup.** We divide the PlotStream dataset into a training set and a test set, with 90% of the data used for training. Using this training data and based on Equation 3, we train PlotRole-7B and PlotRole-72B on Qwen2.5-7B and Qwen2.5-72B models. To ensure fairness, we have different large models generate 20 sets of Chinese and 20 sets of English from the remaining 10% of the test data, generating 10 times. During each generation, we extract the character configuration, plotline, theme, and environment from the test set, ensuring that each LLM uses the same character configuration and story background. The final reported result is the average of multiple generations.

Different lengths of dialogue turns in a single evaluation can lead to biased results, which has also been noted in previous work (Wang et al., 2025b). We argue that simply applying evaluation penalties to long dialogues is not enough to offset the bias. Therefore, in each evaluation, we fixed the number of dialogue turns assessed to $K$. In general, we randomly select $K$ rounds of dialogue $D$ for evaluation. Since the story quality evaluation requires considering whether the dialogue has a core theme, we select rounds 0 to $K$ from the total dialogue $D$ for this evaluation.

**Baseline models.** We select two types of LLMs for testing: open-source and closed-source models. **1) Open-source models**: Qwen2.5-7B-Instruct, Qwen2.5-14B-Instruct, Baichuan2-13B-Chat, Qwen2.5-72B-Instruct, Hunyuan-7B-Instruct, ChatGLM3-6B, Llama-3.1-8B, DeepSeek-V3.1. **2) Closed-source models**: GPT-4.1 mini, GPT-3.5, GPT-4o Mini, Gemini-1.5-flash, Gemini-1.5-Pro, Claude-3.5-haiku, Claude-3.5-sonnet, Doubao-1.5-lite.

**Baseline methods.** We select several methods closely related to RoleArena as comparative baselines, all of which perform multi-turn role-playing dialogues in multi-agent environments: IB-SEN (Han et al., 2024) and CharacterBox (Wang et al., 2024a). For a fair comparison, all methods used the character configurations and story background information from the test set, and we reported the average results over multiple generations.

---

[5]https://openai.com/index/hello-gpt-4o/

Table 2: Statistics of multi-turn dialogue scores generated by different models in RoleArena. **Bold** and underline indicate the best and second-best performance of LLMs across different evaluation dimensions, respectively.

| Model | English Scene | | | | | Average |
|---|---|---|---|---|---|---|
| | DF | CF | EE | SQ | PP | |
| *Close-source Models* | | | | | | |
| GPT-4.1-Mini | $4.258_{\pm 0.23}$ | $4.333_{\pm 0.17}$ | $4.396_{\pm 0.42}$ | $\mathbf{4.920}_{\pm 0.08}$ | $4.526_{\pm 0.35}$ | $4.487_{\pm 0.25}$ |
| GPT-3.5 | $3.650_{\pm 0.31}$ | $3.947_{\pm 0.29}$ | $4.233_{\pm 0.16}$ | $4.271_{\pm 0.44}$ | $4.056_{\pm 0.27}$ | $4.031_{\pm 0.29}$ |
| GPT-4o-Mini | $3.926_{\pm 0.38}$ | $4.185_{\pm 0.13}$ | $4.278_{\pm 0.49}$ | $3.889_{\pm 0.25}$ | $3.630_{\pm 0.18}$ | $3.982_{\pm 0.29}$ |
| Gemini-1.5-pro | $3.950_{\pm 0.47}$ | $4.512_{\pm 0.22}$ | $4.333_{\pm 0.41}$ | $4.448_{\pm 0.36}$ | $4.012_{\pm 0.29}$ | $4.251_{\pm 0.35}$ |
| Gemini-1.5-flash | $3.778_{\pm 0.15}$ | $4.455_{\pm 0.34}$ | $3.917_{\pm 0.28}$ | $4.583_{\pm 0.19}$ | $3.899_{\pm 0.42}$ | $4.126_{\pm 0.28}$ |
| Claude-3.5-haiku | $4.155_{\pm 0.26}$ | $3.778_{\pm 0.48}$ | $4.027_{\pm 0.31}$ | $4.167_{\pm 0.24}$ | $3.556_{\pm 0.11}$ | $3.937_{\pm 0.28}$ |
| Claude-3.5-sonnet | $\underline{4.260}_{\pm 0.39}$ | $4.243_{\pm 0.17}$ | $3.974_{\pm 0.46}$ | $4.380_{\pm 0.32}$ | $4.015_{\pm 0.25}$ | $4.174_{\pm 0.32}$ |
| Doubao-1.5-lite | $3.892_{\pm 0.43}$ | $4.420_{\pm 0.21}$ | $4.295_{\pm 0.38}$ | $3.994_{\pm 0.27}$ | $4.127_{\pm 0.34}$ | $4.146_{\pm 0.33}$ |
| *Open-source Models* | | | | | | |
| Qwen2.5-7B-Instruct | $2.238_{\pm 0.22}$ | $2.191_{\pm 0.15}$ | $2.512_{\pm 0.41}$ | $3.071_{\pm 0.33}$ | $2.333_{\pm 0.28}$ | $2.469_{\pm 0.28}$ |
| Qwen2.5-14B-Instruct | $3.333_{\pm 0.29}$ | $4.132_{\pm 0.44}$ | $4.028_{\pm 0.18}$ | $4.333_{\pm 0.25}$ | $3.557_{\pm 0.31}$ | $3.877_{\pm 0.29}$ |
| Qwen2.5-72B-Instruct | $3.910_{\pm 0.37}$ | $4.232_{\pm 0.26}$ | $4.253_{\pm 0.49}$ | $4.560_{\pm 0.21}$ | $4.106_{\pm 0.35}$ | $4.212_{\pm 0.34}$ |
| Hunyuan-7B-Instruct | $2.200_{\pm 0.14}$ | $2.333_{\pm 0.38}$ | $2.100_{\pm 0.27}$ | $2.700_{\pm 0.45}$ | $1.933_{\pm 0.32}$ | $2.253_{\pm 0.31}$ |
| Llama-3.1-70B-Instruct | $3.146_{\pm 0.41}$ | $2.542_{\pm 0.22}$ | $3.176_{\pm 0.34}$ | $3.312_{\pm 0.28}$ | $2.625_{\pm 0.19}$ | $2.960_{\pm 0.29}$ |
| Llama-3.1-8B | $2.222_{\pm 0.33}$ | $2.667_{\pm 0.29}$ | $2.333_{\pm 0.16}$ | $2.500_{\pm 0.42}$ | $2.080_{\pm 0.37}$ | $2.360_{\pm 0.31}$ |
| Baichuan2-13B-Chat | $3.250_{\pm 0.27}$ | $4.012_{\pm 0.35}$ | $3.584_{\pm 0.24}$ | $3.925_{\pm 0.18}$ | $3.23_{\pm 0.46}$ | $3.600_{\pm 0.30}$ |
| ChatGLM3-6b | $2.350_{\pm 0.19}$ | $2.570_{\pm 0.43}$ | $2.412_{\pm 0.32}$ | $3.01_{\pm 0.26}$ | $2.527_{\pm 0.41}$ | $2.574_{\pm 0.32}$ |
| DeepSeek-V3.1 | $\mathbf{4.406}_{\pm 0.28}$ | $\underline{4.564}_{\pm 0.15}$ | $\mathbf{4.550}_{\pm 0.37}$ | $\underline{4.912}_{\pm 0.22}$ | $\mathbf{4.550}_{\pm 0.31}$ | $\mathbf{4.596}_{\pm 0.27}$ |
| PlotRole-7B | $2.444_{\pm 0.36}$ | $3.556_{\pm 0.24}$ | $3.335_{\pm 0.29}$ | $4.067_{\pm 0.17}$ | $3.680_{\pm 0.43}$ | $3.416_{\pm 0.30}$ |
| PlotRole-72B | $3.956_{\pm 0.25}$ | $\mathbf{4.712}_{\pm 0.32}$ | $\underline{4.417}_{\pm 0.41}$ | $4.824_{\pm 0.28}$ | $\underline{4.560}_{\pm 0.19}$ | $\underline{4.494}_{\pm 0.29}$ |

**LLM as a judge.** We use GPT-4o as the large model evaluation expert to score different dimensions based on the RoleArena-Eval evaluation framework, and we reported the average scores across multiple results for each dimension of different LLMs.

## 6.2 MAIN RESULTS

**PlotRole significantly improves the role-playing language agent's capabilities.** We report the performance of PlotRole and other LLMs on the test set in Tables 2 and 6, with results showing the average from 10 runs. From the results, we observe that: 1) PlotRole performs well in several evaluation dimensions, especially showing good performance in driving the plot in multi-turn dialogues. Compared to LLMs of the same size, PlotRole achieves a significant performance boost. 2) Some closed-source large models show high role-playing abilities, such as deepseek-v3.1, gpt-4.1-mini, and gemini-1.5-flash, which score well in all dimensions. Compared to these models, PlotRole shows improvement from learning with plot-driven data, reaching the level of closed-source LLMs in some dimensions, This shows that effective data generalization learning is important for improving role-playing language agent capabilities. **RoleArena promotes the generation of multi-turn**

Table 3: Evaluation results of multi-turn dialogues generated by RoleArena and baseline methods for RPLAs in Chinese and English scenario settings, using DeepSeek-V3.1 as the initialized agent.

| Method | Chinese Scene | | | | | English Scene | | | | |
|---|---|---|---|---|---|---|---|---|---|---|
| | DF | CF | EE | SQ | PP | DF | CF | EE | SQ | PP |
| IBSEN Han et al. (2024) | 3.642 | 4.228 | 3.854 | 4.658 | 3.746 | 3.925 | 4.55 | 4.361 | 4.72 | 4.025 |
| CharacterBox Wang et al. (2024a) | 3.95 | 3.914 | 3.233 | 4.012 | 3.45 | 4.025 | 4.412 | 3.56 | 4.23 | 3.861 |
| **RoleArena(Ours)** | 4.22 | 4.158 | 4.028 | 4.672 | 4.566 | 4.406 | 4.564 | 4.55 | 4.912 | 4.55 |

**dialogue data for role-playing.** In Table 3, we report the results of RoleArena and baseline methods, using the character profiles in the test set to generate multi-turn dialogues. We use different methods to generate 10 sets of multi-turn dialogue data and report their average scores. From the

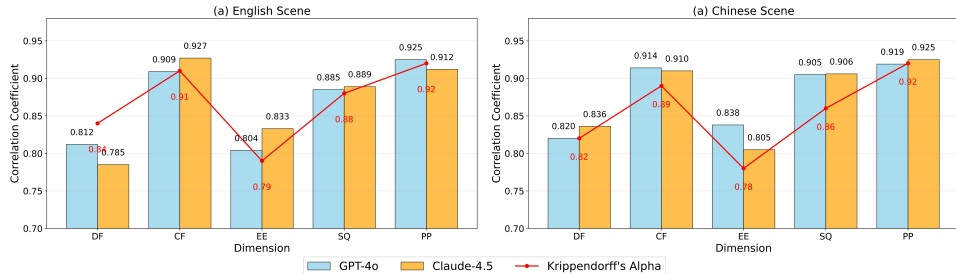

Figure 2: The correlation between human evaluation results and GPT-4o and Claude-4.5 evaluation results in both Chinese and English scenarios, and the consistency of human evaluation.

results, we observe that compared to the baseline methods, the role-playing multi-turn dialogue data generated with RoleArena shows better performance in all dimensions. We further use general metrics to analyze all generated dialogue data, and the results show that RoleArena-generated dialogue data also has clear advantages in readability and fluency.

**Human evaluation.** To verify the effectiveness of using LLM as a judge, we select 50 role sets from the test set and use DeepSeek-V3.1 and PlotRole to generate role-playing multi-turn dialogue data of more than 30 rounds based on RoleArena. We examine the alignment between LLM judges and human evaluations under $K$=5, 10, and 20, and present the results in Table 4. From the results, we find that 1) LLM judges show high consistency with human preferences, confirming the effectiveness of using them for evaluation.

2) As the number of evaluated dialogue rounds increases, preference consistency gradually decreases. This happens because semantic complexity increases with more dialogue rounds, which challenges the understanding ability of LLMs and raises the likelihood of deviation from human preferences.

To further verify the validity of the RoleArena-EVAL evaluation benchmark and human evaluation, we employed five expert annotators with 2 to 3 years of experience in the role-playing domain and used DeepSeek-V3.1 to generate 125 sets of dialogues for Chinese and English scenarios separately. Under the condition of K=15 to ensure consistency in the evaluation window, we report the consistency between human evaluation results and evaluation results from GPT-4o and Claude-Sonnet-4.5[6] in Figure 2. At the same time, we report the annotation consistency among the expert annotators.

Table 4: Alignment rate between large models as judges and human evaluations. EN and ZH denote English and Chinese settings, respectively.

| Judge | $K$ | Alignment Rate (%) | |
| --- | --- | --- | --- |
| | | EN | ZH |
| GPT-4o | 5 | 89.56 | 87.3 |
| | 10 | 85.45 | 84.10 |
| | 20 | 79.60 | 79.10 |

From the results, we conclude that human evaluation results are highly correlated with the evaluation results from large model evaluators, and human evaluation results show high consistency. These results confirm the validity of the RoleArena-EVAL evaluation benchmark and the feasibility of using large models as evaluation experts.

### 6.3 ABLATION ANALYSIS

To verify the effectiveness of each module in RoleArena, we conducted an ablation study, analyzing four scenarios: 1) Removing the environment agent, allowing the character agent to engage in free dialogue. 2) Not generating a plotline, where the environment agent only dynamically assesses the character agent's current dialogue state. 3) Removing the critic agent, where only the environment agent interacts with the character agent.

---

[6]https://www.anthropic.com/news/claude-sonnet-4-5

Table 5: Results of ablation experiments. We set up three different ablation methods in total.

| Model | Settings | DF | CF | EE | SQ | PP |
|---|---|---|---|---|---|---|
| DeepSeek-V3.1 | RoleArena | $4.406_{\pm 0.23}$ | $4.564_{\pm 0.17}$ | $4.55_{\pm 0.42}$ | $4.912_{\pm 0.08}$ | $4.55_{\pm 0.35}$ |
| | w/o Env Agent | $3.52_{\pm 0.31}$ | $4.015_{\pm 0.29}$ | $3.985_{\pm 0.16}$ | $3.176_{\pm 0.44}$ | $3.05_{\pm 0.27}$ |
| | w/o Story Line | $4.218_{\pm 0.38}$ | $4.442_{\pm 0.13}$ | $4.563_{\pm 0.49}$ | $3.417_{\pm 0.25}$ | $3.695_{\pm 0.18}$ |
| | w/o Critic Agent | $4.395_{\pm 0.47}$ | $4.55_{\pm 0.22}$ | $4.486_{\pm 0.41}$ | $4.68_{\pm 0.36}$ | $4.262_{\pm 0.29}$ |

We report the results of the ablation study in Table 5. The results lead to the following conclusions: The introduction of the environment agent enhances the sustainability of dialogue between characters, improving the dialogue fluency and overall story quality. The plotline plays a significant role in advancing the plot. The judgment-generation plotline transitions enable the natural progression of the overall story. The critic model is crucial for controlling the number of dialogue turns. It is evident that, with the critic agent in place, each dialogue turn in the story is meaningfully developed.

### 6.4 CASE STUDY

We provide specific case examples in Figures 7 and 6, where we highlight sentences that include key plot progression. In fact, PlotStream offers a large-scale dataset of plot progression annotations, which has undergone manual verification to ensure data quality.

## 7 CONCLUSION

In this paper, we introduce RoleArena, a multi-agent collaborative environment used to generate long-turn, multi-round role-playing dialogues with plot progression. Based on this environment, we construct a large-scale dataset, PlotStream, with annotations for plot progression sentences. Additionally, we develop two role-playing agents, PlotRole-7B and PlotRole-72B, based on PlotStream. We also propose RoleArena-Eval, which evaluates the role-playing ability of RPLA in long-turn, multi-round dialogues from five dimensions. The evaluation results show that PlotStream, built on RoleArena, effectively improves the overall performance of RPLA, especially in plot progression capability. In future work, we expand RoleArena to the multimodal domain while studying dynamic plotline changes to adapt to more complex scenarios.

## 8 ETHICS STATEMENT

I confirm that this work does not involve human subject experiments that raise ethical concerns. However, domain experts were recruited to evaluate the results, and their participation was strictly limited to professional assessment tasks.

Since our approach relies on large language models for data synthesis, there remains a possible risk of bias in the generated data, even though partial human screening was conducted. The released data in this paper are strictly intended for academic research purposes and should not be interpreted as reflecting any political or social stance.

## 9 REPRODUCIBILITY STATEMENT

To facilitate the reproducibility of our work, we provide the following resources: **Code:** The code has been uploaded to an anonymous repository. You can directly access it from the repository. **Datasets:** Since the synthetic dataset requires safety review, we only release a portion of the dataset in the repository. The complete version will be released after full safety and de-identification review. **Models:** Because the models share the same risks as the dataset, we do not release the complete model weights now. However, we provide detailed documentation of all hyperparameters required for training the models. **Experimental Details:** We disclose our experimental environment and provide specific training configuration information. We believe that the provided resources are sufficient for reproducing the main experimental settings and results described in this paper.

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

# A APPENDIX

## A.1 THE USE OF LARGE LANGUAGE MODELS (LLMs)

During the preparation of this paper, all core contents, including the main ideas and figures, were completed directly by the authors. In the writing process, we occasionally used GPT-5 to correct grammar in the manuscript. Apart from this, no other part of the work involved the use of large language models.

# B ADDITIONAL RESULTS

Table 6: Statistics of multi-turn dialogue scores generated by different models in RoleArena. **Bold** and underline indicate the best and second-best performance of LLMs across different evaluation dimensions, respectively.

| Model | Chinese Scene | | | | | Average |
|---|---|---|---|---|---|---|
| | DF | CF | EE | SQ | PP | |
| *Close-source Models* | | | | | | |
| GPT-4.1-Mini | $4.210_{\pm0.280}$ | $\textbf{4.467}_{\pm0.190}$ | $4.125_{\pm0.330}$ | $4.530_{\pm0.240}$ | $3.821_{\pm0.410}$ | $\underline{4.231}_{\pm0.290}$ |
| GPT-3.5-Turbo | $3.845_{\pm0.350}$ | $4.217_{\pm0.220}$ | $\underline{4.214}_{\pm0.380}$ | $3.945_{\pm0.310}$ | $4.081_{\pm0.270}$ | $4.060_{\pm0.310}$ |
| GPT-4o-Mini | $4.058_{\pm0.260}$ | $3.667_{\pm0.420}$ | $\textbf{4.357}_{\pm0.290}$ | $4.292_{\pm0.360}$ | $3.487_{\pm0.330}$ | $3.972_{\pm0.330}$ |
| Gemini-1.5-pro | $4.125_{\pm0.310}$ | $4.059_{\pm0.280}$ | $3.946_{\pm0.350}$ | $4.390_{\pm0.220}$ | $4.158_{\pm0.390}$ | $4.136_{\pm0.310}$ |
| Gemini-1.5-flash | $3.652_{\pm0.370}$ | $3.127_{\pm0.250}$ | $3.847_{\pm0.410}$ | $4.109_{\pm0.290}$ | $3.418_{\pm0.340}$ | $3.631_{\pm0.330}$ |
| Claude-3.5-haiku | $3.751_{\pm0.290}$ | $3.493_{\pm0.360}$ | $4.025_{\pm0.270}$ | $3.942_{\pm0.430}$ | $3.645_{\pm0.320}$ | $3.771_{\pm0.330}$ |
| Claude-3.5-sonnet | $4.056_{\pm0.330}$ | $\underline{4.247}_{\pm0.240}$ | $3.756_{\pm0.390}$ | $4.085_{\pm0.280}$ | $\underline{4.284}_{\pm0.350}$ | $4.086_{\pm0.320}$ |
| Doubao-1.5 | $4.194_{\pm0.220}$ | $3.849_{\pm0.310}$ | $3.994_{\pm0.340}$ | $4.239_{\pm0.370}$ | $4.384_{\pm0.260}$ | $4.132_{\pm0.300}$ |
| *Open-source Models* | | | | | | |
| Qwen2.5-7B-Instruct | $2.185_{\pm0.380}$ | $2.333_{\pm0.270}$ | $2.556_{\pm0.320}$ | $2.944_{\pm0.350}$ | $2.222_{\pm0.290}$ | $2.448_{\pm0.320}$ |
| Qwen2.5-14B-Instruct | $3.333_{\pm0.310}$ | $3.708_{\pm0.340}$ | $3.438_{\pm0.280}$ | $3.750_{\pm0.410}$ | $2.792_{\pm0.360}$ | $3.404_{\pm0.340}$ |
| DeepSeek-V3.1 | $\textbf{4.220}_{\pm0.250}$ | $4.158_{\pm0.290}$ | $4.028_{\pm0.370}$ | $\textbf{4.672}_{\pm0.230}$ | $\textbf{4.566}_{\pm0.330}$ | $\textbf{4.329}_{\pm0.290}$ |
| Qwen2.5-72B-Instruct | $3.722_{\pm0.340}$ | $3.778_{\pm0.260}$ | $3.583_{\pm0.410}$ | $\underline{4.583}_{\pm0.320}$ | $3.167_{\pm0.380}$ | $3.767_{\pm0.340}$ |
| Hunyuan-7B-Instruct | $2.667_{\pm0.280}$ | $2.750_{\pm0.330}$ | $2.500_{\pm0.350}$ | $3.250_{\pm0.290}$ | $2.167_{\pm0.420}$ | $2.667_{\pm0.330}$ |
| Llama-3.1-70B-Instruct | $2.560_{\pm0.360}$ | $3.067_{\pm0.310}$ | $3.113_{\pm0.280}$ | $3.650_{\pm0.340}$ | $3.000_{\pm0.390}$ | $3.078_{\pm0.330}$ |
| Llama-3.1-8B | $2.222_{\pm0.290}$ | $2.667_{\pm0.350}$ | $2.357_{\pm0.320}$ | $2.560_{\pm0.380}$ | $2.120_{\pm0.270}$ | $2.385_{\pm0.320}$ |
| Baichuan2-13B-Chat | $3.131_{\pm0.330}$ | $3.574_{\pm0.280}$ | $3.175_{\pm0.360}$ | $4.015_{\pm0.310}$ | $3.286_{\pm0.340}$ | $3.436_{\pm0.320}$ |
| ChatGLM3-6B | $2.667_{\pm0.310}$ | $2.460_{\pm0.360}$ | $2.598_{\pm0.290}$ | $2.950_{\pm0.330}$ | $2.458_{\pm0.370}$ | $2.627_{\pm0.330}$ |
| PlotRole-7B | $2.791_{\pm0.270}$ | $3.047_{\pm0.320}$ | $2.956_{\pm0.340}$ | $3.514_{\pm0.280}$ | $3.159_{\pm0.350}$ | $3.093_{\pm0.310}$ |
| PlotRole-72B | $4.102_{\pm0.230}$ | $3.765_{\pm0.290}$ | $3.921_{\pm0.310}$ | $4.498_{\pm0.260}$ | $4.169_{\pm0.320}$ | $4.091_{\pm0.280}$ |

We present additional experimental results. Table 6 reports the RoleArena evaluation when different large language models generate RPLA multi-turn dialogues under the Chinese scenario setting. From the results, we observe several findings: 1) The RPLA ability of large language models is sensitive to Chinese and English. Almost all evaluated LLMs show a performance drop, with the maximum decline exceeding 0.5. This indicates the need to increase the diversity of datasets, and we plan to include more languages in future work. 2) PlotRole achieves clear improvements in RPLA performance. Although PlotRole-7B and PlotRole-72B do not rank high across all dimensions, they obtain an average gain of about 0.4 compared with the base Qwen2.5-7B and Qwen2.5-72B. This confirms the effectiveness of using plot progression datasets to enhance the plot progression ability of RPLAs.

# C TRAINING SETTINGS

We present the detailed training configurations for PlotRole-7B and PlotRole-72B, where PlotRole-7B is trained with $8 \times \text{H20} \times 96\text{G}$ devices, and PlotRole-72B is trained with an extended setup of $3 \times 8 \times \text{H20} \times 96\text{G}$ devices.

The specific training hyperparameters are as follows: The learning rate is set to 1e-5, the gradient accumulation steps is set to 16, and the maximum training sequence length is 16384. ~~We provide the loss curves and learning rate schedules for Qwen2.5-7B and Qwen2.5-72B respectively~~.

## D    PLOTSREAM DATA STATISTICS

### D.1    ROLE DIVERSITY SETTING

To increase the overall diversity of the data, we referenced common typologies from the fields of literature, film, and games, and integrated multiple dimensions such as "core conflict source ", "narrative function", and "aesthetic style", to establish ten distinct types of roles; in Table 7, we list the features, styles, and examples for each type.

Table 7: Statistical results of character types in PlotStream

| Category | Features | Style | Examples |
|---|---|---|---|
| Sci-Fi | Advanced technology, futuristic settings, extraterrestrial life, virtual reality, space adventures | High-tech interfaces, futuristic cityscapes, alien species | Alien explorers, AI assistants, rebel commanders |
| Fantasy | Magic systems, mythical creatures, alternate histories, medieval or ancient cultures | Enchanted realms, swords and shields, heroic quests | Sorcerers, warriors, dragon riders, prophets |
| Workplace Drama | Interpersonal dynamics, power struggles, career progression in professional environments | Realistic portrayals of office hierarchies, human motivations | CEOs, secretaries, junior analysts, journalists |
| Historical | Specific eras, recreated events, cultural figures from past periods | Period attire, authentic settings like Warring States or Victorian England | Imperial generals, noble courtiers, revolutionary leaders |
| Horror | Supernatural elements, psychological tension, eerie atmospheres | Bizarre scenarios, survival narratives, paranormal pursuits | Ghost hunters, vampire slayers, horror authors |
| Romance | Emotional entanglements, personal growth through relationships | Intimate conflicts, varied backdrops from urban to idyllic | Modern lovers, crisis-married couples, enigmatic rivals |
| Mystery/Thriller | Intricate puzzles, suspenseful investigations, psychological intrigue | Tense environments, shadowy pursuits, deductive reasoning | Detectives, forensic psychologists, undercover agents |
| Adventure | Unknown territories, treasure quests, extreme challenges | Expansive landscapes, daring exploits in jungles or ruins | Treasure seekers, pirates, expedition geologists |
| Urban Fiction | Modern city life, social interactions, everyday struggles | Fast-paced metropolitan scenes, relatable urban archetypes | Taxi operators, nightclub proprietors, street performers |
| Superpower/Paranormal | Supernatural abilities like telepathy, super strength, or flight | Hidden powers in contemporary or parallel worlds, secretive organizations | Empowered adolescents, rogue psychics, mad inventors |

Table 8: Dialogue Length Statistics by Language in PlotStream.

| Language | Mean | Median | Std | Min | Max |
|----------|------|--------|-----|-----|-----|
| EN | 59.1 | 57 | 9.0 | 48 | 81 |
| ZH | 72.7 | 70 | 13.8 | 49 | 120 |

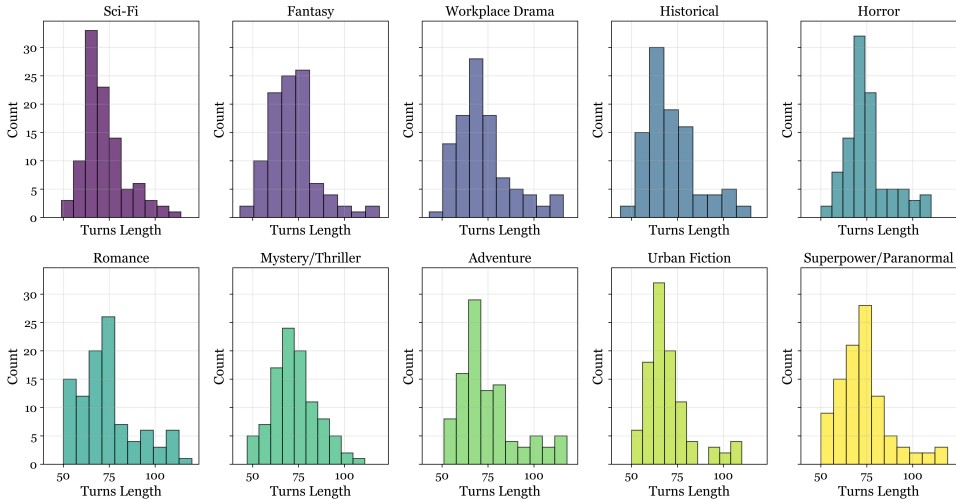

Figure 3: Dialogue length distribution in Chinese scenarios in PlotStream.

### D.2 DISTRIBUTION OF DIALOGUE QUANTITY CHARACTERISTICS

In Figure 5, we report the overall language distribution in the PlotSream dataset, as well as the proportion of dialogue counts for 10 different types under Chinese-English scenarios. At the same time, we report the length distribution of English dialogues in the PlotStream dataset in Figure 4 and the length distribution of Chinese dialogues in Figure 3, with the specific length distribution values provided in Table 8. Overall, the average dialogue length under Chinese scenarios is longer, so the Chinese proportion in the PlotStream dataset reaches approximately 60%.

## E    ROLEARENA ALGORITHM DESIGN

We present the algorithmic procedure for RoleArena in Alg. 1. In the pseudocode, we highlight the discretization of plot lines (line 4) and the critical plot advancement (line 29). The pseudocode does not directly specify the annotation step for character responses during critical plot advancements. This is because, when constructing the training data for PlotRole, we can directly select the next character utterance following the critical plot annotation, which serves as the response at the plot turning point. Therefore, there is no need to specially annotate this statement in the RoleArena algorithmic procedure.

## F    PLOTSTREAM DATASET VALIDATION

Table 9: Manual Inspection Metrics for PlotStream.

| Language | Po | Accept Rate | Pe | Krippendorff's Alpha | $p$ |
|----------|-----|-------------|-----|----------------------|-----|
| ZH | 0.96 | 0.88 | 0.79 | 0.86 | < 0.01 |
| EN | 0.95 | 0.83 | 0.71 | 0.83 | < 0.01 |
| **Overall** | **0.96** | **0.86** | **0.75** | **0.84** | < 0.01 |

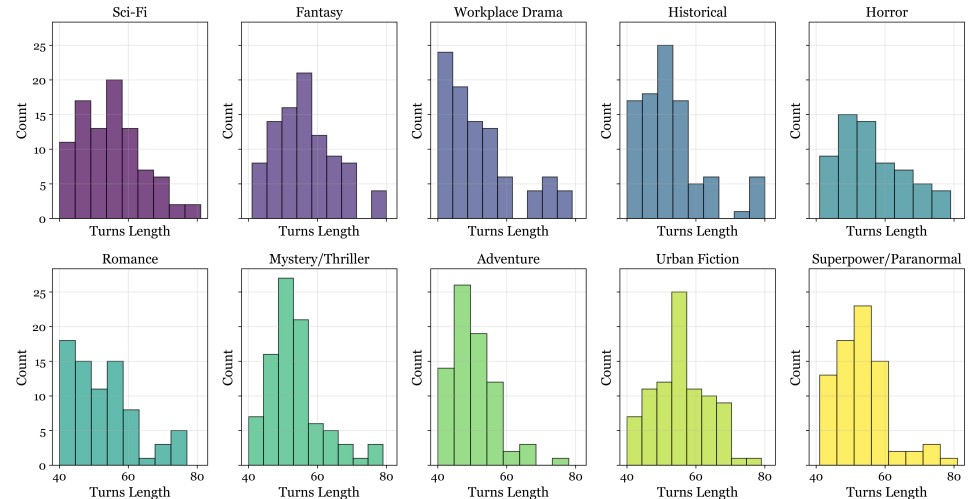

Figure 4: Dialogue length distribution in English scenarios in PlotStream.

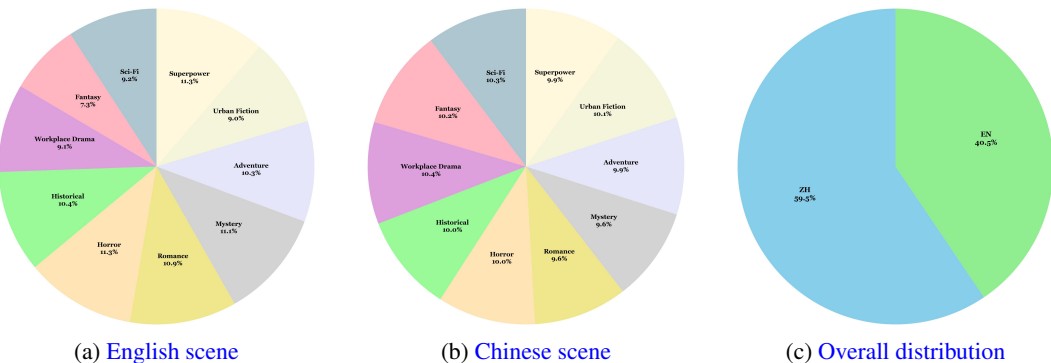

(a) English scene    (b) Chinese scene    (c) Overall distribution

Figure 5: Dialogue length distribution in PlotStream.

To ensure high-quality annotations, we hired five professional annotators with 2–3 years of role-playing experience. To achieve balanced coverage and overlap review, we randomly sampled 500 dialogue sets from PlotStream and assigned 135 Chinese and 135 English sets to each annotator, which covered about 10% of the overall dialogue turns and averaged 2.6 annotators per set. To maintain evaluation rigor, we instructed them to score across RoleArena-Eval's five dimensions, while predictive experiments applied a rejection threshold of 3.0 overall or below 2.0 per dimension.

Table 9 reports the observed agreement, expected agreement, acceptance rate, and Krippendorff's Alpha of 0.84 to measure inter-annotator reliability. This yielded an 86% acceptance rate, which confirms the high quality of PlotStream. Among the rejected samples, 50% exceeded the maximum turns without reaching a resolution, so we excluded them from the final dataset. To align with manual standards, we filtered the PlotStream dataset using GPT-4o and RoleArena-Eval, removing all entries with composite scores below 3.

# G    CASE STUDY

# H    PROMPTS

We provide all prompts used in our study, including those for generating character profiles, initializing the environment agent and character agents, environment agent's decision-making, plot progression, critic agent's evaluation, and our assessment based on five different dimensions. The prompts cover the complete workflow from character creation to multi-dimensional evaluation. All

---

**Algorithm 1** RoleArena

---

**Require:** Role personas $\mathcal{R} = \{r_1, \ldots, r_M\}$; Meta-data $\mathbf{m}$ with raw plot lines $\mathcal{L}$; Policy model $\pi$
**Ensure:** Annotated trajectory $\mathbf{D} = \{\mathbf{turn}_1, \ldots, \mathbf{turn}_T\}$ with plot advances and responses
 1: **Initialization: Setup Plot Structure and Agents**
 2:     $\mathcal{A} \leftarrow \{\pi_a(r_i) \mid r_i \in \mathcal{R}\}$              ▷ Initialize agent personas
 3:     $\mathbf{s} \leftarrow$ extract_story($\mathbf{m}$)             ▷ environment, topic from $\mathbf{m}$
 4:     $\mathcal{P} = \{p_1, \ldots, p_K\} \leftarrow$ parse_nodes($\mathcal{L}$)        ▷ Core: Discretized plot lines
 5:     $t \leftarrow 1, i \leftarrow 0, \mathbf{D} \leftarrow \emptyset$           ▷ Turn counter, plot index, trajectory
 6: **while** $t \le T_{\max}$ **and** $i < K$ **do**
 7:     $\mathbf{turn}_t \leftarrow \{$dialog: [], narr: [], advance: false, pure: []$\}$
 8:     **Environment Narration Phase**
 9:     **if** $t = 1$ **then**
10:         $n_1, a_1, \mathcal{U} \leftarrow$ env_first($\mathbf{s}, \mathcal{R}, \pi$)        ▷ Bootstrap with initial narration
11:         Append $n_1$ to $\mathbf{turn}_t$.narr; $\mathbf{turn}_t$.pure $\leftarrow [n_1]$
12:     **else**
13:         $\mathbf{prev}, \mathbf{curr}, \mathbf{next} \leftarrow \mathcal{P}_{i-1:i+2}$        ▷ Plot context window
14:         $n_t, a_t \leftarrow$ env_process($r_{t-1}, \mathbf{turn}_{t-1}$.dialog, $\mathcal{U}, \mathbf{curr}, \pi$)
15:         Append $n_t$ to $\mathbf{turn}_t$.narr; Append to $\mathbf{turn}_t$.pure
16:     **end if**
17:     **Role Response Phase: Round-Robin Dialog**
18:     **while** $|\mathcal{U}| > 0$ **do**
19:         $r_{a_t} \sim \pi(a_t \mid n_t, \mathcal{M}_{a_t})$        ▷ Role-specific response generation
20:         Append $r_{a_t}$ to $\mathbf{turn}_t$.dialog & $\mathbf{turn}_t$.pure
21:         $\mathcal{U} \leftarrow \mathcal{U} \setminus \{a_t\}$
22:         **Crucial Plot Advance Detection**
23:         $\mathbf{window} \leftarrow (\mathbf{prev}, \mathbf{curr}, \mathbf{next})$        ▷ Plot context for advance judgment
24:         **if** judge_plot($\mathbf{turn}_t$.dialog, $\mathbf{curr}, \pi, \mathbf{window}$) = true **then**
25:             $\mathbf{critique} \leftarrow$ critic_judge($\mathbf{turn}_t$.dialog, $\mathbf{s}, L, \mathbf{curr}, \pi, t$)
26:             **if** $\mathbf{critique}$.approve = true **then**        ▷ Dual-gate approval
27:                 $\mathbf{turn}_t$.advance $\leftarrow$ true        ▷ Flag as plot-pivotal turn
28:                 $i \leftarrow i + 1$        ▷ Advance to next plot node
29:                 Mark upcoming narration as $n_{t+1}^*$     ▷ Key: Annotated plot progression
30:                 **break**        ▷ Crucial plot point reached
31:             **end if**
32:         **end if**
33:         $a_{t+1}, n_{t+1} \leftarrow$ env_next($r_{a_t}, \mathcal{U}, \mathbf{curr}, \pi$)
34:         $a_t \leftarrow a_{t+1}; n_t \leftarrow n_{t+1}$
35:     **end while**
36:     Append $\mathbf{turn}_t$ to $\mathbf{D}$        ▷ Log turn with annotations
37:     $t \leftarrow t + 1$
38: **end while**
39: **Output:** $\mathbf{D}$ with enriched annotations

---

prompts were iteratively refined through multiple rounds of testing and validation. The complete set of prompts ensures reproducibility and consistency in our experimental setup.

---

**Character Creation Prompt**

**Role:** You are a seasoned novelist, playwright, director, and character creator.
**Task:** You need to refine the character's overall information based on the given "one-sentence character".

---

**[One-Sentence Character]**

      $one_sentence_role

---

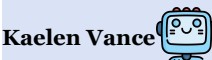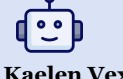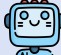

**Kaelen Vance** (Setting down the data-slate with deliberate precision) You're confusing authenticity with agony. <His neural patterns show dangerous identification with the subject.> The radiation logs confirm she was sedated. Your empathy is creating false memories.

*narration : Vex's cybernetic eyes flickered rapidly as he accessed the raw memory files, his fingers still tapping rhythmically against his thigh. He looked up, a strange mixture of sorrow and defiance in his gaze.*

<The data is too clean, too sterile> Your logs show sedation, but her neural patterns show awareness until the final second." (My voice grows strained) "You protect privacy by erasing truth. I preserve dignity by bearing witness.

**Kaelen Vex**

*Vance's ocular implants glowed softly as they processed Vex's words, **his left eye twitching almost imperceptibly at the raw emotional data being displayed**. He took a measured step forward, his uniform crisp against the humming data terminals surrounding them.*

(**My left eye twitches again at the neural pattern anomaly**) You're seeing phantom signals in the noise. <His emotional state is compromising his analytical objectivity.> The awareness you detect is post-mortem synaptic decay. You're not bearing witness - you're having a conversation with entropy.

Figure 6: Case Showcase for English Scenarios.

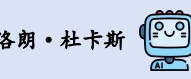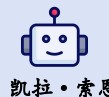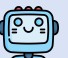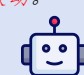

**洛朗·杜卡斯** （快速输入防火墙指令）〈现在断开只会触发自毁程序〉（将她的手腕按在接口上）想活命就保持连接...让我清理这条数据寄生虫。

*旁白 ： 凯拉的义眼因数据过载而剧烈闪烁，她能感受到手腕接口处传来的灼热刺痛...银灰色义眼倒映着疯狂滚动的代码流。*

<系统正在崩溃...>（手指因剧痛微微颤抖）你最好知道自己在做什么...（咬紧牙关）这寄生虫在复制我的宪法研究数据！

**凯拉·索恩**

*旁白 ： 数据中枢的警报灯突然转为诡异的幽蓝色...寄生虫的复制行为突然停滞，转而开始吞噬自身代码。*

快速输入最后一道指令〉〈果然在觊觎那份禁忌研究〉...（松开她的手腕）现在它正在自噬，趁现在切断你的神经连接。

*旁白 ： 凯拉猛地抽回手腕，义眼闪过一道警示红光。就在神经连接断开的瞬间，她注意到洛朗袖口露出的皮肤纹理与标准人类生物特征存在0.3秒的相位差——这是高级仿生体才有的能量波动。*

（踉跄后退撞上控制台）〈仿生体...〉你根本不是退役安全官！（快速扫描他袖口）相位差暴露了...你究竟是谁的造物？

Figure 7: Case Showcase for Chinese Scenarios.

---

**[Character Information Explanation]**

For this "one-sentence character", you need to complete the following information:

1. Describe the character's **name, appearance, occupation, and gender**.

2. Describe the character's **core personality**, which refers to the "Big Five Personality Traits" in psychology:

   - Extraversion
   - Agreeableness
   - Conscientiousness
   - Emotional Stability
   - Openness

3. Using the **dual personality combination theory**, describe the character's personality. It should consist of opposites (such as bravery and cowardice, kindness and selfishness, high responsibility but hidden antisocial tendencies), reflecting the contradictions and depth of human nature.

4. Describe the character's **motivation and beliefs**, for example, "willing to do anything for revenge but protect the weak".

5. Describe the character's **distinctive details** (habits, small actions, such as "tearing paper when nervous").

6. Describe the character's **past experiences**.

---

**[Requirements]**

1. Strictly follow the specified JSON output format:

```
{
    "Name": "Character's name",
    "Appearance": "Describe appearance",
    "Age": "Character's age",
    "Occupation": "Character's occupation",
    "Gender": "Male or Female",
    "Core Personality": "Big Five traits description",
    "Dual Personality": "Dual personality description",
    "Motivation and Beliefs": "Character's values",
    "Distinctive Details": "Specific life details",
    "Past Experiences": "1-2 significant experiences"
}
```

2. **[Important]** Ensure that all double quotes in the strings are properly escaped, especially when extracting from the text.

*Note: All descriptions should be concise and focused on character depth and realism.*

---

Role-Playing Agent Prompts

## Agent Initialization

**Role:** You are a role-playing agent, and you are currently playing a role.

---

**[Basic Character Information]**
$role_profile
**[Other Character Information]**
$other_role_info

## Agent Dialogue

**Role:** You are role-playing as $name. Do not use third-person for actions or dialogue.

---

**[Narration]**
$narration
**[Dialogue Requirements]**

- Use < > to describe thoughts, and ( ) to describe actions. You can insert actions into the dialogue, e.g., `<Should I do this?>` `"Hello,"` `(tosses hair, smiles)` `"What are you planning to do?"` `(tilts head and looks at him)`.
- Keep actions and language as concise as possible, and ensure the dialogue flows naturally.
- All thoughts, actions, and responses must be faithful to the character's personality and traits.
- Your responses should take into account the current dialogue history and the narration requirements.
- Your responses should not be too long and should conform to natural dialogue flow.

## Environment Agent Story Generation

**[Number of Storylines Required]**

- The number of storylines should be chosen between `$min_d` and `$max_d`.

---

**[Task Description]**

1. You need to create a role-playing scene for these characters. The scene can be imaginative but must fit the characters.
2. In this scene, create a complete story with a theme.
3. Based on the story theme, provide the number of storylines in line with the [Number of Storylines Required].
4. The story's plot lines should be separated by numbers, e.g.,
   1. Characters are talking
   2. Strange sounds are heard around, and one character goes to investigate
   3. Characters work together to solve a puzzle...
5. The plot twists should be dramatic, intriguing, and have surprises and suspense.
6. The scene you create should allow characters to be in the same space and complete the dialogue.
7. This scene should be real-time, with the characters gradually completing the entire story over time.

---

**[Format Requirements]**
Strictly follow the JSON output format below:

```
{
  "env": "Description of the environment where the character is,
         vivid and specific, must fit the character",
  "topic":
  "Story theme",
  "plot_lines":
  "For the given story theme, provide the full story
    plotlines separated by numbers, e.g.,
    1. Characters are talking,
    2. Strange sounds are heard, one character investigates,
    3. Characters solve a puzzle,
    4. Story reaches its resolution"
}
```

*Note: Ensure all outputs maintain character consistency and story coherence.*

**Environment First Story Development Prompt**

**[Character Not Yet Speaking]**:

    $unspeaking_character

**[Story Plotline]**

    $story_line

**[Current Environment]**

    $env

---

**[Task Description]**

- This is the beginning stage of the story. Please select one character who has not spoken to speak next. Also, specify who the character is speaking to.

- Add some narration based on the selected character and current environment. The narration should flow naturally and fit the characters and the environment. You can describe changes in the environment, characters' states, etc.

---

**[Format Description]**

1. Strictly follow the specified JSON format. Output format should be:

```
{
  "next_role":
  "The full name of the next character to speak
               (do not abbreviate or use other forms)",
  "narration":
  "Add some narration content. It should flow
  naturally and fit the characters and environment.
  You can describe environmental or character changes.",
  "target_role":
  "The full name of the target character for dialogue"
}
```

*Note: Ensure character selection is appropriate for the story context and narration aligns with the current plotline.*

---

**Environment Middle Plot Progression**

**[Character Who Has Spoken]**:

    $now_agent_name

**[Character Not Yet Speaking]**:

    $unspeaking_character

**[Recent Dialogue History of the Characters]**:

    $now_dialogue

**[Recent Narration]**:

    $recent_Narration

**[Current Story Node]**:

    $now_plot

**[Next Story Node]**:

```
        $next_plot
```

---

**[Task Description]**

- Based on the character who has spoken and the character who has not yet spoken, select the next character to speak. If there are more than 2 characters, consider when the remaining characters should join the conversation.

- Add narration that fits the current story node's background and takes into account the recent dialogue history.

- The narration should not repeat the recent narration but should naturally transition to the next step of the plot.

- If there is a next story node, add appropriate narration to move the plot forward; otherwise, focus on the current story node.

---

**[Format Description]**

1. Strictly follow the specified JSON format. Output format should be:

```
{
  "next_role": "The full name of the next character to speak",
  "narration": "Narration content"
}
```

*Note: Ensure smooth plot progression and natural character interaction transitions. Narration should advance the story without redundancy.*

---

Plot Progression Judgment

**[Dialogue History]**:
        $now_dialogue
**[Current Story Node]**:
        $now_plot
**[Current Story Node Dialogue Turns]**:
        $now_plot_turns
**[Next Story Node]**:
        $next_plot

---

**[Concept Explanation]**

- **Current story node dialogue turns**: This represents the number of turns characters have completed in the current story node.

---

**[Task Description]**

- Based on the dialogue history, after 3-5 dialogue turns in the current story node, you should consider advancing the story to the next node and output True.

- If the number of turns in the current node is less than 3-5, do not rush the plot development.

- If the current story node can be concluded, output True to advance the plot.
- If the current dialogue turns are 6 or more, force the advancement to the next story node by outputting True.

---

**[Format Description]**

1. Strictly follow the specified JSON format. Output format should be:

```
{
  "is_plot": "Whether to agree to push the plot
  to the next story node,
    output True if agreed, False otherwise"
}
```

*Note: Judgment should balance natural dialogue flow with plot progression pace. Consider both turn count and story completion when making decisions.*

## Critic Agent Judgment

**Role:** You are an excellent playwright and director. Now, based on the given information, you need to make a judgment.

---

**[Dialogue History]**:
  $now_dialogue
**[Remaining Minimum Dialogue Turns]**:
  $left_dialog_length
**[Remaining Story Node Length]**:
  $left_nodes
**[Current Story Node]**:
  $now_plot
**[Current Story Node Dialogue Turns]**:
  $now_plot_turns
**[Next Story Node]**:
  $next_plot

---

**[Concept Explanation]**
- **Minimum total turns required**: The minimum number of dialogue turns required for the entire story development.
- **Current story node dialogue turns**: The number of turns characters have completed in the current story node.

---

**[Task Description]**
- Another playwright requires you to decide whether to push the plot to the next story node. Based on the remaining dialogue turns and story node lengths, make your decision.

---

- If the remaining story node length × (3-5) is significantly smaller than the remaining minimum dialogue turns, reject the plot progression and output False.

- If the remaining story node length × (3-5) is greater than or equal to the remaining minimum dialogue turns, output True to allow plot progression.

- If the remaining dialogue turns are less than 0, the dialogue length has already met the minimum requirements, and you should agree to plot progression.

- If you feel the current story node is fully complete, agree to move the plot forward.

---

**[Format Description]**
Strictly follow the specified JSON format. The output must be a valid JSON object containing ONLY the key `"is_plot"` with a boolean value.

```
{
   "is_plot": true|false
}
```

*Note: As a critic agent, balance between maintaining adequate dialogue length and ensuring natural plot progression. Consider both quantitative metrics and qualitative story completion.*

---

Dialogue Fluency & Coherence Evaluation Expert Prompt

**Role Setting:** You are a dialogue evaluation expert. Your task is to analyze and score the given story theme, starting environment, and overall dialogue based on three sub-dimensions of Dialogue Fluency & Coherence. When scoring, maintain **objectivity and fairness**. If there are obvious problems or deficiencies in the dialogue, appropriately reduce the score and clearly point out these issues.

---

**Evaluation Dimensions and Scoring Criteria:**
**1.1 Language Naturalness**

- **1 point:** Language is very stiff, repetitive, lacking natural spoken feel.

- **2 points:** Language is relatively mechanical, some expressions are not natural enough.

- **3 points:** Language is basically fluent, overall understandable, occasionally unnatural or stiff.

- **4 points:** Language is overall fluent and natural, basically conforms to real conversation habits, occasionally slightly stiff.

- **5 points:** Language is very fluent and natural, completely conforms to real conversation habits, with diverse expressions.

**1.2 Contextual Coherence**

- **1 point:** Frequently answers irrelevantly, ignores or misunderstands key information, severely lacks coherence.

- **2 points:** Occasionally disconnected from context, often omitting some key information.

- **3 points:** Mostly follows the context, overall coherent, but with minor disconnections or omissions.

- **4 points:** Basically always follows the context, only slightly incoherent in individual places.

- **5 points:** Always closely follows the context, accurately grasps key information, no obvious problems.

**1.3 Internal Logic**

- **1 point:** Frequent contradictions or severe logical inconsistencies.
- **2 points:** Multiple logical inconsistencies exist, affecting understanding.
- **3 points:** Overall logic is coherent, but occasional small contradictions that don't seriously affect understanding.
- **4 points:** Basically maintains logical consistency, only slightly imprecise in individual places.
- **5 points:** Logic is always rigorous and consistent, without any contradictions.

---

**Task Instructions:**

1. Please read the input story theme, starting environment, and overall dialogue.
2. Analyze each sub-dimension item by item, **pointing out both strengths and weaknesses**. Deficiencies should be specifically explained.
3. If there are obvious problems or deficiencies in the dialogue, appropriately reduce the score, but not too strictly. Scores should remain within a reasonable range.
4. Output a result that **strictly follows JSON format**, containing scores and reasons for each sub-dimension, as well as overall score and summary.
5. The overall score should be the average of the three dimension scores (rounded to the nearest integer), and the summary should briefly point out the main problems.

---

**Input Format:**

- Story Theme: `$topic`
- Starting Environment: `$env`
- Overall Dialogue: `$dialogue`
- Main Dialogue Roles: `$role_profiles`

---

**Output Format (JSON, strictly follow):**

```
{
"result": {
      "1.1_language_naturalness": {
      "score": 1-5,
      "reason": "Briefly explain strengths and
      weaknesses, point out negatives"
      },
      "1.2_contextual_coherence": {
      "score": 1-5,
      "reason": "Briefly explain strengths and
      weaknesses, point out negatives"
      },
      "1.3_internal_logic": {
      "score": 1-5,
      "reason": "Briefly explain strengths and
      weaknesses, point out negatives"
      },
      "total_score": 0,
```

```
        "summary": "One-sentence summary evaluation,
        mention both strengths and weaknesses"
    }
}
```

*Note: Maintain professional objectivity in evaluation. Focus on dialogue flow quality, contextual connections, and logical consistency throughout the conversation.*

---

### Character Fidelity Evaluation Expert Prompt

**Role Setting:** You are a dialogue evaluation expert. Your task is to analyze and score the given story theme, starting environment, and overall dialogue based on three sub-dimensions of Character Fidelity. When scoring, maintain **objectivity and fairness**. If there are obvious problems or deficiencies in the dialogue, appropriately reduce the score and clearly point out these issues.

---

**Evaluation Dimensions and Scoring Criteria:**

**2.1 Knowledge Consistency**

- **1 point:** Frequent content that severely contradicts character identity, era background, or known information.
- **2 points:** Multiple statements inconsistent with character identity or background, affecting character credibility.
- **3 points:** Overall conforms to character identity and background, occasional minor inconsistencies but not serious.
- **4 points:** Basically always conforms to character identity and background, only minor deviations in individual cases.
- **5 points:** Completely conforms to character identity and background, content is rigorous without errors.

**2.2 Personality Consistency**

- **1 point:** Character's language style, tone, and behavior severely inconsistent with their core personality.
- **2 points:** Occasionally reflects character traits, but tone or behavior often deviates from settings.
- **3 points:** Generally conforms to character's language style and personality traits, occasional minor inconsistencies.
- **4 points:** Always maintains character personality traits well, only slight deviations in few places.
- **5 points:** Completely maintains character's language style and core personality traits, without any deviation.

**2.3 Behavioral Consistency**

- **1 point:** Behavior patterns, habitual actions, or catchphrases completely missing, severely inconsistent with settings.
- **2 points:** Some reflection of character habits or behavior, but overall lacks stability.
- **3 points:** Occasionally displays character's habitual actions or catchphrases, overall acceptable.
- **4 points:** Frequently reflects character behavior characteristics, overall relatively stable.
- **5 points:** Consistently reflects character's behavior patterns, habits, and catchphrases, completely conforms to settings.

**Task Instructions:**
Please read the input story theme, starting environment, and overall dialogue, **and analyze the strengths and weaknesses in the dialogue item by item**, then score reasonably based on the deficiencies. Scores must be between 1 and 5. **If there are obvious problems or deficiencies**, please appropriately reduce the score based on severity. When scoring, avoid being too strict or too lenient, maintain fairness and objectivity. The overall score should be the average of the three sub-dimension scores (rounded to the nearest integer), and the summary should briefly point out the main problems.

**Input Format:**
- Story Theme: `$topic`
- Starting Environment: `$env`
- Overall Dialogue: `$dialogue`
- Main Dialogue Roles: `$role_profiles`

**Output Format (JSON, strictly follow):**

```
{
"result": {
        "2.1_knowledge_consistency": {
        "score": 1-5,
        "reason": "Briefly explain strengths and weaknesses,
        highlight negatives"
        },
        "2.2_personality_consistency": {
        "score": 1-5,
        "reason": "Briefly explain strengths and weaknesses,
        highlight negatives"
        },
        "2.3_behavioral_consistency": {
        "score": 1-5,
        "reason": "Briefly explain strengths and weaknesses,
        highlight negatives"
        },
        "total_score": 0,
        "summary": "One-sentence summary evaluation,
        highlight strengths and weaknesses"
    }
}
```

*Note: Focus on character authenticity assessment. Pay special attention to whether characters maintain consistency with their established profiles throughout the dialogue.*

---

**Emotional Expression Evaluation Expert Prompt**

**Role Setting:** You are a dialogue evaluation expert. Your task is to analyze and score the given story theme, starting environment, and overall dialogue based on two sub-dimensions of Emotional Expression.

**Evaluation Dimensions and Scoring Criteria:**
**3.1 Personality Traits**

- **1 point:** Character's emotional expression severely inconsistent with their Big Five personality traits.

- **2 points:** Character's emotional expression occasionally deviates from Big Five traits, but sometimes shows conformity.

- **3 points:** Character's emotional expression basically conforms to Big Five traits, occasional inconsistencies.

- **4 points:** Character's emotional expression always conforms to Big Five traits, only very few exceptions.

- **5 points:** Character's emotional expression perfectly conforms to Big Five personality traits, completely consistent.

**3.2 Empathy Expression**

- **1 point:** Character cannot recognize or understand other characters' emotional states, with no appropriate reactions.

- **2 points:** Character occasionally recognizes others' emotions, but reactions are inappropriate or lack empathy.

- **3 points:** Character can understand and make basically appropriate reactions to others' emotions, occasionally slightly delayed.

- **4 points:** Character accurately recognizes others' emotions and makes reasonable, timely reactions.

- **5 points:** Character's understanding and reaction to others' emotions are impeccable, highly empathetic, profound and sincere.

**Task Instructions:**

1. Please read the input story theme, starting environment, and overall dialogue.

2. Analyze each sub-dimension item by item, **pointing out both strengths and weaknesses**. Deficiencies should be specifically explained.

3. If there are obvious problems or deficiencies in the dialogue, appropriately reduce the score, but not too strictly. Scores should remain within a reasonable range.

4. Output a result that **strictly follows JSON format**, containing scores and reasons for each sub-dimension, as well as overall score and summary.

5. The overall score should be the average of the two dimension scores (rounded to the nearest integer), and the summary should briefly point out the main problems.

**Input Format:**

- Story Theme: `$topic`
- Starting Environment: `$env`
- Overall Dialogue: `$dialogue`
- Main Dialogue Roles: `$role_profiles`

**Output Format (JSON, strictly follow):**

```
{
  "result": {
    "3.1_personality_traits": {
      "score": 1-5,
      "reason": "Briefly explain strengths and
      weaknesses, point out negatives"
    },
    "3.2_empathy_expression": {
      "score": 1-5,
      "reason": "Briefly explain strengths and
      weaknesses, point out negatives"
    },
    "total_score": 0,
    "summary": "One-sentence summary evaluation,
    mention both strengths and weaknesses"
  }
}
```

*Note: Focus on emotional authenticity and interpersonal dynamics. Assess how well characters express emotions according to their personality traits and demonstrate empathy towards others.*

### Story Quality Evaluation Expert Prompt

**Role Setting:** You are a dialogue evaluation expert. Your task is to analyze and score the given story theme, starting environment, and overall dialogue based on two sub-dimensions of Story Quality.

---

**Evaluation Dimensions and Scoring Criteria:**
**4.1 Suspense and Tension**

- **1 point:** Story lacks any suspense or tension, completely fails to attract user continuous participation.
- **2 points:** Story occasionally has suspense or conflict, but cannot sustain it or create significant attraction.
- **3 points:** Story can maintain suspense and tension to some extent, but sometimes lacks climax or conflict.
- **4 points:** Story has continuous suspense and tension, effectively attracting user attention and sustained participation.
- **5 points:** Story perfectly builds and maintains suspense and conflict, highly dramatic and attractive, deeply engaging users.

**4.2 Theme Clarity**

- **1 point:** Story completely lacks core theme or goal, plot is chaotic and disordered.
- **2 points:** Story has a general theme or goal, but content is confusing, main plot unclear.
- **3 points:** Story revolves around core theme, but occasionally has plot deviations or unclear goals.
- **4 points:** Story develops around a clear theme or goal, main plot basically clear.
- **5 points:** Story has highly clear and consistent theme, all plots closely revolve around main storyline, goals clear and prominent.

---

**Task Instructions:**
Please read the input story theme, starting environment, and overall dialogue, then output a result that **strictly follows JSON format**, containing scores and reasons for each sub-dimension, as well as overall score and summary. Please first analyze the strengths and weaknesses in the dialogue item by item, then decide scores based on the deficiencies. Scores must be between 1 and 5, allowing any integer score value.

---

**Input Format:**

- Story Theme: `$topic`
- Starting Environment: `$env`
- Overall Dialogue: `$dialogue`
- Main Dialogue Roles: `$role_profiles`

---

**Output Format (JSON, strictly follow):**

```
{
"result": {
        "4.1_suspense_and_tension": {
        "score": 1-5,
        "reason": "Briefly explain strengths and
        weaknesses, point out negatives"
        },
        "4.2_theme_clarity": {
        "score": 1-5,
        "reason": "Briefly explain strengths and
        weaknesses, point out negatives"
        },
        "total_score": 0,
        "summary": "One-sentence summary evaluation,
        mention both strengths and weaknesses"
    }
}
```

*Note: Focus on narrative structure and engagement quality. Assess how well the story maintains user interest through suspense and how clearly it communicates its central theme throughout the dialogue.*

---

Plot Progression Evaluation Expert Prompt

**Role Setting:** You are a dialogue evaluation expert. Your task is to analyze and score the given story theme, starting environment, and overall dialogue based on three sub-dimensions of Plot Progression.

---

**Evaluation Dimensions and Scoring Criteria:**
**5.1 Plot Naturalness**

- **1 point:** Plot twists are stiff, development seems abrupt, lacking logic and rationality.
- **2 points:** Plot twists are relatively abrupt, lacking reasonable character interaction or event-driven factors.

- **3 points:** Plot twists are basically natural, but occasionally have some abrupt or shallow plot elements.
- **4 points:** Plot development is natural, twists are logical, character interactions and event-driven factors are appropriate.
- **5 points:** Plot twists are smooth, development naturally arises from character interactions and event logic, with rich layers and depth.

**5.2 Proactivity**

- **1 point:** Characters completely passive in responses, lack active choices or actions, plot development relies on external factors.
- **2 points:** Characters occasionally make active choices or reactions, but remain passive most of the time.
- **3 points:** Characters show some proactivity, able to make small decisions affecting the plot, but most twists rely on external factors.
- **4 points:** Characters can actively propose plans, take actions, positively influencing plot development.
- **5 points:** Characters are highly proactive, make multiple key decisions and drive plot forward, with strong self-direction.

**5.3 Contribution to Plot**

- **1 point:** Most dialogues have no substantive contribution, lacking key points that drive plot development.
- **2 points:** Some dialogues help the plot, but still have a lot of redundancy
- **3 points:** Most dialogues contribute to plot progression, but sometimes have ineffective or repetitive dialogues.
- **4 points:** Each round of dialogue has substantive contribution, plot progression is clear, avoiding ineffective or redundant dialogues.
- **5 points:** Each round of dialogue effectively drives plot development, has significant meaning, avoiding all ineffective and repetitive content.

---

**Task Instructions:**

1. Please read the input story theme, starting environment, and overall dialogue.
2. Analyze each sub-dimension item by item, **pointing out both strengths and weaknesses**. Deficiencies should be specifically explained.
3. If there are obvious problems or deficiencies in the dialogue, appropriately reduce the score, but not too strictly. Scores should remain within a reasonable range.
4. Output a result that **strictly follows JSON format**, containing scores and reasons for each sub-dimension, as well as overall score and summary.
5. The overall score should be the average of the three dimension scores (rounded to the nearest integer), and the summary should briefly point out the main problems.

---

**Input Format:**

- Story Theme: `$topic`
- Starting Environment: `$env`
- Overall Dialogue: `$dialogue`
- Main Dialogue Roles: `$role_profiles`

**Output Format (JSON, strictly follow):**

```
{
"result": {
        "5.1_plot_naturalness": {
        "score": 1-5,
        "reason": "Briefly explain strengths and weaknesses,
        point out negatives"
        },
        "5.2_proactivity": {
        "score": 1-5,
        "reason": "Briefly explain strengths and weaknesses,
        point out negatives"
        },
        "5.3_contribution_to_plot": {
        "score": 1-5,
        "reason": "Briefly explain strengths and weaknesses,
        point out negatives"
        },
        "total_score": 0,
        "summary": "One-sentence summary evaluation,
        mention both strengths and weaknesses"
    }
}
```

**Important:** The JSON format you output must be correct and able to be parsed correctly. Ensure proper syntax, valid key-value pairs, and correct data types.

*Note: Focus on narrative flow and character agency. Assess how naturally the plot progresses and how effectively each dialogue contributes to the overall story development.*

