# OpenReview forum: "RoleArena: A Multi-Agent Role-Playing Environment for Long Multi-Turn Dialogues with Autonomous Plot Progression"
_ICLR.cc/2026/Conference — Submitted to ICLR 2026_

### Official Review · Reviewer_8Dqz · 2025-10-30

**Soundness:** 2
**Presentation:** 2
**Contribution:** 2
**Rating:** 4
**Confidence:** 4

**Summary:**

This paper introduces RoleArena, a multi-agent simulation environment designed to generate long, plot-driven multi-turn dialogues for role-playing language agents (RPLAs). It proposes an integrated framework that combines environment agents, critic agents, and character agents to achieve controllable plot progression. The authors also release PlotStream, a large-scale dataset (39.3k characters, 170k dialogues), and train two RPLAs, PlotRole-7B and PlotRole-72B, evaluated through the proposed RoleArena-Eval benchmark.

**Strengths:**

1. Clear Motivation and Problem Definition: The paper correctly identifies a gap in RPLA research: the lack of datasets and environments emphasizing plot progression in long multi-turn dialogues.
2. Comprehensive Evaluation: The proposed RoleArena-Eval provides a well-defined five-dimensional metric system (Dialogue Fluency, Character Fidelity, Emotional Expression, Story Quality, Plot Progression).

**Weaknesses:**

1. Limited Novelty of Framework: RoleArena builds directly upon the design principles of prior systems like CharacterBox, COSER, and RolePlot, extending them mainly by adding a critic module. The conceptual leap, while practical, may be viewed as an engineering improvement rather than a methodological innovation.
2. Evaluation Bias (LLM-as-Judge): Heavy reliance on GPT-4o as an evaluator raises questions about LLM bias, even though a human evaluation subsection is provided. It would strengthen the paper to include human–LLM agreement statistics beyond Table 4 (e.g., Pearson correlation coefficient, Cohen’s Kappa).
3. Limited Evaluation Coverage: The evaluation omits several recent high-performing closed-source models such as Gemini-2.5 and Claude-4, even though they were released before DeepSeek-V3.1. Instead, only older versions were tested. This incomplete coverage limits the credibility of the comparative analysis. Moreover, if those newer models show stronger dialogue and narrative abilities, they might serve as better candidates than DeepSeek-V3.1 for constructing higher-quality PlotStream data.
4. Dataset Quality Control: The dataset appears largely synthetic, generated via DeepSeek-V3.1 and other LLMs, with “partial human screening.” It would be beneficial to provide a detailed explanation of the quality control strategies that have been adopted.

**Questions:**

1. When the publications are not included in the sentence, the citations should be in parentheses using \citep{}.
2. Does the paper have detailed statistics or examples for the PlotStream dataset, such as average dialogue length, language ratio, and plot type distribution? Including such analyses or sample dialogues could help clarify the dataset’s quality and diversity.
3. Are there evaluation results for Qwen2.5-7B-Instruct and Qwen2.5-72B (base) models? Was PlotRole trained on the base models or the instruct variants of Qwen2.5? If it used the base models, what was the reason for not choosing the instruct versions, which might be better aligned for dialogue generation tasks?
4. Does RoleArena suffer from plot stagnation, where dialogues drift or repeat without reaching the point for plot progression?

---

> ### Author Response · Authors · 2025-11-23
> **Response to Reviewer 8Dqz (1/4)**
>
> Thank you very much for taking the time to review our paper and for providing high-quality comments. We highly value your feedback, and we will now provide detailed responses to each of your questions and concerns.
>
> ## W'1: Addressing Questions on the Framework's Innovativeness
>
> We greatly appreciate your valuable suggestions and also thank you for recognizing the practicality of RoleArena. We will now provide a clear explanation of the innovation of RoleArena.
>
> #### Overall Innovation
>
> 1. **Innovative direction**. To the best of our knowledge, we are the first to study plot progression in long-turn role-playing multi-turn dialogues, as reflected in your feedback as well as in the comments from reviewers *1Bbq* and *GrBA*.
> 2. **Innovative framework**. RoleArena does not add modules to existing frameworks, but introduces the critic agent and designs an innovative multi-agent dynamic interaction framework to enable long-turn autonomous plot progression in role-playing dialogues. In the next section, we will provide a more detailed analysis of the differences from existing frameworks.
> 3. **Filling a gap in the field**. Our work fills a critical gap in the role-playing field by providing long-turn multi-turn dialogue data with plot progression.
>
> #### Framework Innovation
>
> RoleArena constructs discrete storylines through the Env Agent, dynamically combines the dialogue history of the Character Agent to make judgments for plot progression, and provides behavioral guidance to the Character Agent. At the same time, RoleArena introduces the critic Agent, achieving long-turn controllable role-playing multi-turn dialogues with built-in plot progression. This "generation + dynamic judgment + consensus-driven" framework paradigm fundamentally differs in design principles from [1] CharacterBox, [2] COSER, and [3] RolePlot.
>
> Additionally, we would like to point out that the plot progression statements in RolePlot rely entirely on the "LLM filtering + manual filtering judgment" framework, which incurs significant labor costs. RoleArena overcomes this limitation through autonomous plot progression and automatic annotation (see the algorithm in **Appendix E**), making it innovative from a framework design perspective.
>
> *[1] Wang L, Lian J, Huang Y, et al. Characterbox: Evaluating the role-playing capabilities of llms in text-based virtual worlds[C]//Proceedings of the 2025 Conference of the Nations of the Americas Chapter of the Association for Computational Linguistics: Human Language Technologies (Volume 1: Long Papers). 2025: 6372-6391.*
>
> *[2] Wang X, Wang H, Zhang Y, et al. Coser: Coordinating llm-based persona simulation of established roles[J]. arXiv preprint arXiv:2502.09082, 2025.*
>
> *[3] Zhang P, An S, Qiao L, et al. RolePlot: A Systematic Framework for Evaluating and Enhancing the Plot-Progression Capabilities of Role-Playing Agents[C]//Proceedings of the 63rd Annual Meeting of the Association for Computational Linguistics (Volume 1: Long Papers). 2025: 12337-12354.*

---

> ### Author Response · Authors · 2025-11-23
> **Response to Reviewer 8Dqz (2/4)**
>
> ## W'2：Evaluation Bias (LLM-as-Judge)
>
> We fully understand your concerns regarding the over-reliance on GPT-4o for evaluation. In response to this issue, we expanded the scale of human evaluation by hiring 5 expert annotators. At the same time, we used Claude-Sonnet-4.5 and GPT-4o as large model evaluation experts to assess 125 sets of generated Chinese and English dialogues.
>
> The results in Table 1 confirm that, based on the RoleEval evaluation system, the multi-model expert evaluations are highly correlated with human evaluations, with an average Pearson correlation coefficient greater than 0.85. At the same time, Krippendorff's Alpha confirms the high consistency of human evaluations, reflecting the stability of evaluations based on RoleArena-Eval. More specific descriptions can be found in our response to reviewer *MDvV*.
>
> | Language | Dim      | Pearson (r) GPT-4o | p(GPT-4o) | Pearson (r) Claude-Sonnet-4.5 | p(Claude-Sonnet-4.5) | Krippendorff's Alpha |
> | -------- | -------- | ------------------ | --------- | ----------------------------- | -------------------- | -------------------- |
> | **EN**   | DF       | 0.812              | <0.01     | 0.785                         | <0.01                | 0.84, p<0.01         |
> |          | CF       | 0.909              | <0.01     | 0.927                         | <0.01                | 0.91, p<0.01         |
> |          | EE       | 0.804              | <0.01     | 0.833                         | <0.01                | 0.79, p<0.01         |
> |          | SQ       | 0.885              | <0.01     | 0.889                         | <0.01                | 0.88, p<0.01         |
> |          | PP       | 0.925              | <0.01     | 0.915                         | <0.01                | 0.92, p<0.01         |
> |          | Avg.(EN) | **0.867**          | -         | **0.869**                     | -                    | **0.87**             |
> | **ZH**   | DF       | 0.820              | <0.01     | 0.836                         | <0.01                | 0.82, p<0.01         |
> |          | CF       | 0.914              | <0.01     | 0.91                          | <0.01                | 0.89, p<0.01         |
> |          | EE       | 0.838              | <0.01     | 0.805                         | <0.01                | 0.78, p<0.01         |
> |          | SQ       | 0.905              | <0.01     | 0.906                         | <0.01                | 0.86, p<0.01         |
> |          | PP       | 0.919              | <0.01     | 0.925                         | <0.01                | 0.92, p<0.01         |
> |          | Avg.(ZH) | **0.879**          | -         | **0.876**                     | -                    | **0.85**             |
>
> ***Table 1**. Correlation and Agreement of Evaluation Results*
>
> ## W'3：Limited Evaluation Coverage
>
> Thank you for raising this important issue; we acknowledge its relevance. In response, we tested three advanced closed-source large models, Gemini-2.5, Claude-Sonnet-4, and Claude-Sonnet-4.5, and reported the results in Table 2 for both Chinese and English test scenarios. The results show that Claude-Sonnet-4.5 and Gemini-2.5 achieved the highest scores in the Chinese and English scenarios, respectively. From the overall scores, it appears that Claude-Sonnet-4.5 and Gemini-2.5 are more suitable for data generation.
>
> However, after further analysis, we found that in both the Chinese and English scenarios, responses from individual characters in the Gemini and Claude models were too long, making them unsuitable for dialogue data. Additionally, when overall performance was similar, DeepSeek-V3.1, as an open-source model, provided better cost-effectiveness. We recognize the need to increase the variety of large models used in data generation to reduce model bias. In the future, we plan to incorporate a wider range of models to optimize the PlotStream dataset. Thank you again for your valuable suggestions.
>
> | Language | LLM               | DF    | CF    | EE    | SQ    | PP    | Average   |
> | -------- | ----------------- | ----- | ----- | ----- | ----- | ----- | --------- |
> | **ZH**   | DeepSeek-V3.1     | 4.220 | 4.158 | 4.028 | 4.672 | 4.566 | 4.329     |
> |          | Gemini-2.5        | 4.105 | 4.341 | 4.256 | 4.431 | 4.603 | 4.347     |
> |          | Claude-Sonnet-4   | 3.934 | 4.017 | 4.128 | 4.374 | 4.506 | 4.192     |
> |          | Claude-Sonnet-4.5 | 4.068 | 4.449 | 4.227 | 4.510 | 4.613 | **4.373** |
> | **EN**   | DeepSeek-V3.1     | 4.406 | 4.564 | 4.550 | 4.912 | 4.550 | 4.596     |
> |          | Gemini-2.5        | 4.215 | 4.725 | 4.574 | 4.915 | 4.582 | **4.602** |
> |          | Claude-Sonnet-4   | 4.392 | 4.218 | 4.624 | 4.712 | 4.495 | 4.488     |
> |          | Claude-Sonnet-4.5 | 4.259 | 4.592 | 4.636 | 4.895 | 4.506 | 4.578     |
>
> ***Table 2**. Evaluation Results of High-Performance Closed-Source Models*

---

> ### Author Response · Authors · 2025-11-23
> **Response to Reviewer 8Dqz (3/4)**
>
> ## W'4：Dataset Quality Control
>
> Thank you for pointing out the shortcomings in our PlotStream data quality control; we acknowledge that this issue does exist. The quality control strategy for the PlotStream dataset is described in detail in our response to reviewer *1Bbq*. In Table 3, we report again the observed agreement rate Po, acceptance rate, and expected agreement rate Pe, and finally provide Krippendorff's Alpha as the measure of inter-annotator agreement. The overall acceptance rate of 86% combined with Krippendorff's Alpha of 0.84 confirms the reliability of the PlotStream dataset quality.
>
> |  language   |    Po    | Accept Rate |    Pe    | Krippendorff's Alpha |     p      |
> | :---------: | :------: | :---------: | :------: | :------------------: | :--------: |
> |     ZH      |   0.96   |    0.88     |   0.79   |         0.86         |   ＜0.01   |
> |     EN      |   0.95   |    0.83     |   0.71   |         0.83         |   ＜0.01   |
> | **overall** | **0.96** |  **0.86**   | **0.75** |       **0.84**       | **＜0.01** |
>
> ***Table 3**: Manual Inspection Metrics for PlotStream*
>
> ## Q'1：Addressing Citation Format Issues
>
> Thank you for pointing out the citation format issue in the paper. We have made adjustments to the citation format throughout the revised version.
>
> ## Q'2: Detailed Statistics of the PlotStream Dataset
>
> Thank you very much for your suggestion. To respond to your suggestion, we reported in Table 4 the proportion of Chinese and English in the PlotStream dataset, as well as the distribution of dialogue lengths in both languages.
>
> In addition, in the **revised version**, we added a detailed description of the PlotStream dataset, and you can review the specific information provided in **Appendix D.2**.
>
> | Language  | Mean | Median | Std  | Min  | Max  |
> | :-------: | :--: | :----: | :--: | :--: | :--: |
> | EN(40.5%) | 59.1 |   57   |  9   |  48  |  81  |
> | ZH(59.5%) | 72.7 |   70   | 13.8 |  49  | 120  |
>
> ***Table 4**. Statistical Results of the PlotStream Dataset*

---

> ### Author Response · Authors · 2025-11-23
> **Response to Reviewer 8Dqz (4/4)**
>
> ## Q'3: Qwen Base Model Evaluation Results and Training Base Model
>
> First, we sincerely apologize for the confusion. In the **revised version**, we clarified the evaluation versions of the Qwen series models and **marked them in red**. Both the evaluation and training in the paper used Qwen2.5-7B-Instruct and Qwen2.5-72B-Instruct. To better address your question, we reported the evaluation results of the two base models, Qwen2.5-7B and Qwen2.5-72B, in Table 5. The results show that Qwen2.5-7B-Instruct and Qwen2.5-72B-Instruct performed better overall.
>
> Regarding the reason for this, we agree with your point: the instruction-tuned versions are more suitable for dialogue tasks, and this is also why we chose the instruction versions for fine-tuning.
>
> | Language | LLM                  | DF    | CF    | EE    | SQ    | PP    | Average |
> | -------- | -------------------- | ----- | ----- | ----- | ----- | ----- | ------- |
> | **ZH**   | Qwen2.5-7B-Instruct  | 2.185 | 2.333 | 2.556 | 2.944 | 2.222 | 2.448   |
> |          | Qwen2.5-72B-Instruct | 3.722 | 3.778 | 3.583 | 4.583 | 3.167 | 3.767   |
> | **EN**   | Qwen2.5-7B           | 2.167 | 2.516 | 2.174 | 2.633 | 1.836 | 2.265   |
> |          | Qwen2.5-72B          | 3.349 | 3.728 | 3.417 | 4.059 | 3.254 | 3.561   |
>
> ***Table 5**. Qwen Base and Instruction-Tuned Model Evaluation Results*
>
> ## Q'4: Ability of RoleArena to Maintain Topic and Advance Plot
>
> Thank you for your question. To address your question, we reported in **Table 6** the cross-model sensitivity experiment results of RoleArena in both Chinese and English scenarios. We point out that introducing a Critic Agent and ensuring that it has strong performance allows the success rate of dialogue generation to exceed 90%. Dialogue generation fails when the conversation does not end after reaching the maximum turn limit. We acknowledge that due to hallucination issues in current large models, it is not yet possible to fully ensure that every conversation completes each plot point within the constrained number of turns.
>
> RoleArena controls the topic and character responses through the Env Agent. With the help of discrete plot points and dynamic judgment based on dialogue history, the Env Agent keeps the dialogue aligned with the current plot point and avoids generating information that repeats previous dialogue content.
>
> | Language | OOD LLM               | LLM               | DF    | CF    | EE    | SQ    | PP    | Average   | Success Rate |
> | -------- | --------------------- | ----------------- | ----- | ----- | ----- | ----- | ----- | --------- | ------------ |
> | **ZH**   | -                     | **DeepSeek-V3.1** | 4.220 | 4.158 | 4.028 | 4.672 | 4.566 | **4.329** | 95%          |
> |          | **Gemini-2.5**        |                   | 4.351 | 4.026 | 4.210 | 4.585 | 4.627 | **4.360** | 94%          |
> |          | **Claude-Sonnet-4.5** |                   | 4.229 | 4.165 | 4.108 | 4.694 | 4.487 | **4.337** | 94%          |
> | **EN**   | -                     | **DeepSeek-V3.1** | 4.406 | 4.564 | 4.550 | 4.912 | 4.550 | **4.596** | 93%          |
> |          | **Gemini-2.5**        |                   | 4.159 | 4.338 | 4.439 | 4.754 | 4.564 | **4.451** | 93%          |
> |          | **Claude-Sonnet-4.5** |                   | 4.025 | 4.135 | 4.208 | 4.652 | 4.690 | **4.342** | 94%          |
>
> ***Table 6**. Generalization Results of RoleArena*
>
> ---
>
> **Finally**, we sincerely thank you for the valuable time you spent reviewing our manuscript and for the important suggestions you provided. Thank you for recognizing our research motivation and contributions. We hope that our responses can further improve the quality of the manuscript. We are very willing to continue communicating with you, and if you have any other questions, please feel free to contact us at any time.

---

> > ### Comment · Reviewer_8Dqz · 2025-11-25
> >
> > Thank you for your detailed response, but I still have some concerns regarding innovation and data quality,  thereby keeping my score.
> >
> > 1. What are the fundamental differences between long conversations with plot progression and general conversations with plot progression? The critic idea is common and recognized to improve performance, but I think proposing only a simple critic agent lacks innovation at present.
> >
> > 2. I did not find any strategies for manual quality control in the response to reviewer 1Bbq. The final acceptance rate of 86% indicates that there is a lot of noise in the data. It is necessary to design quality checks to filter out low-quality samples. Additionally, the fact that models like Gemini produce very long outputs in a single round does not mean that they cannot be used for dialogue construction.

---

> > > ### Author Response · Authors · 2025-11-25
> > > **Response to Reviewer 8Dqz**
> > >
> > > In addition, we would like to add one more point: regarding the fact that after removing the data portions exceeding the maximum dialogue turn limit, the remaining acceptance rate reaches **93%**, this can actually be somewhat verified from the generalization results table of RoleArena in Table 6 of our response to you. Table 6 reports the success rates of Chinese-English dialogue generation in the RoleArena test environment as **95%** and **93%**, respectively.
> > >
> > > We sincerely thank you again for your valuable suggestions. We hope to address your concerns and are very willing to communicate further with you.

---

> ### Author Response · Authors · 2025-11-25
> **Response to Reviewer 8Dqz**
>
> We greatly appreciate the reviewer's valuable comments. To address your current concerns, we provide further explanation here.
>
> ### 1) Difference of Long Plot-Progression Dialogues from general Dialogues
>
> In fact, the essential difference between the two lies in the number of dialogue turns. However, as the number of dialogue turns increases, how to better analyze the performance of RPLA and the quality of the overall dialogue in this situation remains a question that needs further research. The motivation for our study on long-turn plot-advancement role-playing multi-turn dialogues has already been elaborated in the paper, and this is also a shortcoming in current research in the RPLA field, which you have already pointed out in the **Strengths** section. Studying long-turn plot-driving role-playing multi-turn dialogues helps to further analyze the consistency and fidelity of role-playing language agents in long-turn role-playing, and it plays an important role in advancing the plot-driving capability of RPLAs.
>
> In addition, we would like to point out that you also noted in the **Strengths** section the lack of datasets in the current RPLA field that emphasize long-turn multi-turn dialogue plot advancement. RoleArena provides a complete solution for this, covering the full process from environment to data to RPLA training.
>
> ### 2) Additional Clarification on Innovation
>
> Thank you for your question. Based on our initial discussion with you regarding innovation, we provide a further elaboration on the innovation here, with a focused explanation on the framework part.
>
> RoleArena does not simply introduce a Critic Agent:
>
> - **Introduction of discrete plot lines and turn-level dynamic plot judgment**. Specifically, RoleArena first constructs a discrete plot line based on the Env Agent. Subsequently, the Env Agent, guided by the discrete plot line, conducts fine-grained turn-level dynamic interactions and plot judgments with the Character based on the current plot line, and then achieves autonomous plot progression in long-turn dialogues after reaching model consensus with Critic Agent.
> - **Plot advancement combining discrete plot lines with Critic Agent's model consensus**. The Critic Agent additionally considers the overall dialogue turns and the discrete plot line when the Env Agent determines plot advancement, ultimately realizing controllable turn-length plot-advancement role-playing multi-turn dialogues.
>
> herefore, from the perspective of the entire framework, there is a substantial difference from existing work. Beyond RoleArena, we further propose the **RoleArena-Eva**l evaluation framework. The experimental results reported in Table 1 confirm that this evaluation framework effectively achieves multi-dimensional assessment of long-turn role-playing multi-turn dialogues.
>
> ### 3) Dataset Manual Quality Control Strategy
>
> Thank you very much for your question. We are sorry that our wording caused you confusion. In our response to reviewer 1Bbq, we provided the personnel conditions for manual quality control of the dataset and the specific evaluation methods.
>
> To clearly explain our control strategy, Table 3 reports the evaluation results after the initial generation of the dataset. Among the rejected samples, 50% of the data were due to incomplete plot lines upon reaching the maximum number of dialogue turns, so we removed this portion of data from the final dataset.
>
> In **Appendix D.2**, we provide the length distribution of the current PlotStream English-Chinese dialogues, which has actually removed the data from failed generations. Therefore, after removing this portion of data, the acceptance rate can be maintained at **93%**. **In addition, using the same rejection threshold as manual evaluation, when finalizing the PlotStream dataset, we used GPT-4o combined with RoleArena-Eval to remove data with composite scores below 3 in the dataset**.
>
> We commit to adding further details on this in the revised version, and based on this, we have supplemented and improved our response to reviewer 1Bbq. Thank you again for your valuable feedback.
>
> ### 4) Feasibility of Building Dialogues with Gemini
>
> We strongly agree with your opinion. In fact, we have already pointed out in our previous response that we plan to include more types of models to optimize the PlotStream dataset. In addition, we would like to explain that when the overall data generation quality is similar, using open-source models that are more cost-effective can help improve the practicality of the method.
>
> ---
>
> Thank you again for providing the opportunity for discussion. We hope our response can address your confusion. If you have any other questions, please feel free to provide them at any time. We are very willing to discuss further with you. Your valuable suggestions have greatly helped us improve our work.

---

### Official Review · Reviewer_GrBA · 2025-10-31

**Soundness:** 3
**Presentation:** 3
**Contribution:** 3
**Rating:** 6
**Confidence:** 4

**Summary:**

This paper proposes RoleArena, a multi-agent framework for generating and evaluating long-horizon role-playing dialogues. The system comprises three large language model agents: an environment agent responsible for plot control, a role agent that performs role-playing, and a commentator (critic) agent that manages pacing and turn balance. These three agents collaborate to advance the plot through a consensus mechanism defined as P(E,C)=1. Based on this framework, the authors construct PlotStream (≈39K characters and 170K dialogues) and train the PlotRole-7B/72B models. They also propose a five-dimensional evaluation system, RoleArena-Eval. Experimental results show that PlotRole substantially improves performance over comparably sized open-source models, achieving results close to some mid-tier closed-source LLMs in certain dimensions, though still trailing behind top models like DeepSeek-V3.1 and GPT-4.1-Mini.

**Strengths:**

1.	The paper precisely identifies a key deficiency in long-horizon role-playing dialogues—the absence of an autonomous plot-progression mechanism—and proposes a coherent and systematic solution to address this challenge.
2.	The introduction of a dedicated critic agent for adaptive control of plot advancement and pacing is conceptually novel. Its effectiveness is demonstrated through ablation experiments.
3.	The proposed PlotStream dataset and RoleArena-Eval benchmark are valuable community resources that provide a much-needed foundation for studying narrative consistency and controlled story generation.
4.	The paper presents comprehensive experiments, including comparisons across 15+ LLMs, module ablation studies, and human–LLM alignment tests, lending strong support to the empirical findings.

**Weaknesses:**

1.	The central formula P(E,C)=1 relies entirely on the semantic agreement between two LLMs, lacking explicit rules or theoretical grounding for when a plot transition should occur. This opaque, black-box mechanism limits transparency and weakens the theoretical rigor of the contribution.
2.	The description of “secondary checks/manual verification” suggests light post hoc inspection rather than systematic annotation. The paper provides no details about the sampling ratio, annotation criteria, or inter-rater consistency (e.g., Cohen’s k). Given that both data generation and annotation are LLM-driven, potential model biases may be embedded in the dataset.
3.	All metrics are judged solely by GPT-4o, which introduces a risk of stylistic or linguistic bias toward GPT-4o’s own preferences. The paper lacks multi-judge validation or variance analysis to ensure ranking stability. Furthermore, it remains unclear how models producing fewer than K turns are handled, possibly biasing results toward verbose models.
4.	The paper compares RoleArena with CharacterBox, but the two frameworks serve distinct purposes: RoleArena enhances generative narrative control, while CharacterBox focuses on evaluating role-playing fidelity. Treating them as direct baselines conflates different objectives, reducing the validity of the comparison.
5.	Certain sections lack clarity—for instance, Section 4.2 should more explicitly distinguish between narrative outputs and character responses in the training objective. In Section 6.2, “roleplot” may be a typographical error for “plotrole.”

**Questions:**

1.	If the environment and critic agents use different base models (or models of differing strengths), how would this affect turn controllability and plot progression stability? Have you conducted cross-model sensitivity experiments?
2.	What is the proportion of data subjected to manual verification in PlotStream, and what were the annotation criteria and consistency metrics? Could the authors provide a comprehensive data quality report, including statistics on linguistic, emotional, and narrative diversity?
3.	Has RoleArena-Eval been tested with alternative evaluators (e.g., Claude, Gemini, DeepSeek) to assess ranking robustness and inter-judge consistency? If not, are there plans to release scripts or datasets enabling multi-judge evaluation?
4.	How are dialogues with fewer than K turns handled during evaluation? Are such samples regenerated or discarded?
5.	If extensive manual verification is required to ensure annotation reliability, how does this approach retain its advantage over traditional datasets that provide manually annotated plot-progression statement?

---

> ### Author Response · Authors · 2025-11-23
> **Response to Reviewer GrBA (1/3)**
>
> We sincerely thank you for your valuable suggestions; these suggestions are very important for improving the completeness and overall quality of our paper, and next, we will provide detailed responses to each of your suggestions.
>
> ## W'1: The Model Consensus Mechanism for P(E,C)=1
>
> Thank you for pointing out the issue of model consensus in RoleArena's plot progression. We acknowledge that the current plot progression requires model consensus between the environment agent and the critic agent. From a methodological perspective, we differ from [1] Han et al. (2024), who base plot judgment on fixed turns, whereas we use LLMs for fine-grained, dynamic plot progression judgments specific to each interaction turn.
>
> Moreover, we introduced the critic agent for model consensus in plot judgment, which improves the robustness of plot progression. In addition, we added rules for forced plot progression to avoid prolonged stagnation in the plot. In our response to reviewer *MDvV*, we mentioned the effectiveness of the forced plot progression rules, and in Table 1, we provide the related results again. From the results, it can be seen that the forced progression rules effectively improve the success rate of dialogue generation.
>
> | Setting            | Max Turns | Turns Range | Success rate | Avg Turns |
> | ------------------ | --------- | ----------- | ------------ | --------- |
> | **base**           | 6         | (3,5)       | 93%          | 4.3       |
> |                    | 5         | (2,4)       | 92%          | 3.5       |
> |                    | 7         | (4,6)       | 93%          | 5.1       |
> | **w/o Constraint** | 6         | (3,5)       | 72%          | 4.1       |
> |                    | 5         | (2,4)       | 78%          | 3.6       |
> |                    | 7         | (4,6)       | 75%          | 5.5       |
>
> ***Table 1**: Sensitivity Analysis of the Threshold for RoleArena Dialogue*
>
> We point out that this model consensus mechanism is scalable; the critic agent can be extended to an agent group, and based on the model consensus within the group, the robustness of the overall framework can be further improved, and we also warmly welcome further improvement work based on RoleArena.
>
> *[1]Han S, Chen L, Lin L M, et al. IBSEN: Director-Actor Agent Collaboration for Controllable and Interactive Drama Script Generation[C]//Proceedings of the 62nd Annual Meeting of the Association for Computational Linguistics (Volume 1: Long Papers). 2024: 1607-1619.*
>
> ## W'2 & Q'2: Addressing Data Quality Checks and Model Bias Concerns
>
> Thank you for highlighting the shortcomings in our PlotStream data quality control and acknowledge this issue. The quality control strategy is detailed in our response to reviewer *1Bbq*. Table 2 reports the observed agreement rate Po, acceptance rate, expected agreement rate Pe, and Krippendorff's Alpha for inter-annotator agreement. The 86% acceptance rate and 0.84 Alpha confirm the dataset's reliability. Among rejected samples, 50% reached the long dialogue limit without ending; we removed these from the final PlotStream dataset to ensure quality.
>
> Additionally, Table 3 provides the correlation between human evaluation results and those from GPT-4o and Claude-4.5. The results show an average Pearson coefficient greater than 0.85 between human and LLM evaluations, demonstrating small overall model bias under our RoleArena-EVAL system.
>
> |  language   |    Po    | Accept Rate |    Pe    | Krippendorff's Alpha |  p-value   |
> | :---------: | :------: | :---------: | :------: | :------------------: | :--------: |
> |     zh      |   0.96   |    0.88     |   0.79   |         0.86         |   ＜0.01   |
> |     en      |   0.95   |    0.83     |   0.71   |         0.83         |   ＜0.01   |
> | **overall** | **0.96** |  **0.86**   | **0.75** |       **0.84**       | **＜0.01** |
>
> ***Table 2**: Manual Inspection Metrics for PlotStream*
>
> *[1]Han S, Chen L, Lin L M, et al. IBSEN: Director-Actor Agent Collaboration for Controllable and Interactive Drama Script Generation[C]//Proceedings of the 62nd Annual Meeting of the Association for Computational Linguistics (Volume 1: Long Papers). 2024: 1607-1619.*

---

> ### Author Response · Authors · 2025-11-23
> **Response to Reviewer GrBA (2/3)**
>
> ## W'3&Q'3&Q'4：On Multi-Model Expert Evaluation and Dialogue Rounds
>
> ### Multi-Model Expert Evaluation
>
> We used **Claude-Sonnet-4.5** and **GPT-4o** as large model evaluation experts and conducted evaluations separately in Chinese and English scenarios. The results in Table 3 confirm that, based on the RoleEval evaluation system, the multi-model expert evaluations are highly correlated with human evaluations, with an average Pearson correlation coefficient greater than 0.85; this reflects the stability of evaluations based on RoleArena-Eval. More specific descriptions can be found in our response to reviewer *MDvV*.
>
> At the same time, we updated the evaluation scripts for multi-expert evaluation in the **anonymous repository mentioned in the paper**; you can access and view them.
>
> | Language | Dim      | Pearson (r) GPT-4o | p(GPT-4o) | Pearson (r) Claude-Sonnet-4.5 | p(Claude-Sonnet-4.5) | Krippendorff's Alpha |
> | -------- | -------- | ------------------ | --------- | ----------------------------- | -------------------- | -------------------- |
> | **EN**   | DF       | 0.812              | <0.01     | 0.785                         | <0.01                | 0.84, p<0.01         |
> |          | CF       | 0.909              | <0.01     | 0.927                         | <0.01                | 0.91, p<0.01         |
> |          | EE       | 0.804              | <0.01     | 0.833                         | <0.01                | 0.79, p<0.01         |
> |          | SQ       | 0.885              | <0.01     | 0.889                         | <0.01                | 0.88, p<0.01         |
> |          | PP       | 0.925              | <0.01     | 0.915                         | <0.01                | 0.92, p<0.01         |
> |          | Avg.(EN) | **0.867**          | -         | **0.869**                     | -                    | **0.87**             |
> | **ZH**   | DF       | 0.820              | <0.01     | 0.836                         | <0.01                | 0.82, p<0.01         |
> |          | CF       | 0.914              | <0.01     | 0.91                          | <0.01                | 0.89, p<0.01         |
> |          | EE       | 0.838              | <0.01     | 0.805                         | <0.01                | 0.78, p<0.01         |
> |          | SQ       | 0.905              | <0.01     | 0.906                         | <0.01                | 0.86, p<0.01         |
> |          | PP       | 0.919              | <0.01     | 0.925                         | <0.01                | 0.92, p<0.01         |
> |          | Avg.(ZH) | **0.879**          | -         | **0.876**                     | -                    | **0.85**             |
>
> ***Table 3**. Correlation and Agreement of Evaluation Results*
>
> ### Validation for Dialogue Turns Less Than K
>
> Because plot points require a certain number of turns to unfold, if K is too small, it will lead to too few plot points, which does not align with our initial intention of studying long-turn multi-turn role-playing dialogues. At the same time, we also need to ensure fairness and consistency in evaluations across different dialogue groups, so in our setting, K is set to 5 to 20.
>
> As mentioned in the paper, considering that RoleArena-EVAL needs to evaluate story quality, during evaluation, K generally starts increasing from the 0th turn. For sample groups where the total number of generated dialogues is less than K, we will regenerate them and use them for the final evaluation.
>
> ## W'4: Rationale for Comparison with the CharacterBox
>
> Thank you for pointing out this issue. We need to point out:
>
> - [2] CharacterBox provides a simulation environment for generating fine-grained character behavior trajectories, and its specific dialogue process is actually an open-ended narrative process.
> - The entire CharacterBox framework also includes the environment, characters, and plot, with sections 3.1 and 3.2 of its paper describing scene creation and autonomous story evolution.
>
> Therefore, for the completeness of the comparison, we include CharacterBox in the baseline methods for comparison. For the rigor of the evaluation, in the actual comparison, besides the plot progression evaluation dimension, we also compared the two methods across the other four dimensions.
>
> From the comparison results in Table 3 of the paper, it can be seen that compared to CharacterBox, RoleArena shows clear improvements in dialogue fluency, story quality, emotional expression, and plot progression.
>
> *[2] Wang L, Lian J, Huang Y, et al. Characterbox: Evaluating the role-playing capabilities of llms in text-based virtual worlds[C]//Proceedings of the 2025 Conference of the Nations of the Americas Chapter of the Association for Computational Linguistics: Human Language Technologies (Volume 1: Long Papers). 2025: 6372-6391.*

---

> ### Author Response · Authors · 2025-11-23
> **Response to Reviewer GrBA (3/3)**
>
> ## W'5: Addressing the Training Objective Explanation
>
> Thank you for your valuable suggestions. In **Section 4.2** , we more clearly distinguish between the narrative target output and the character response target output in the training objectives. Specifically, we define the character response target to represent the specific reactions of the character during key plot progression, including thoughts, actions, and utterances.
>
> In comparison, the narrative target output represents the narration information output by the environment agent during key plot progression. We have added the new content to the **revised version** and **marked it with blue font**.
>
> ## Q'1: Cross-Model Sensitivity Analysis
>
> This is a very **interesting and important question**. To better address your question, we selected two models with a significant difference in parameter scale, DeepSeek-V3.1 and Qwen2.5-7B-Instruct, mentioned in the paper, for cross-model experiments. We generated 100 groups of dialogues in both Chinese and English scenarios and reported the average results in Table 4.
>
> Our results show that: **1)** A high-performance Critic Agent can greatly enhance the success rate of dialogue generation. We analyzed specific samples and found that using a low-performance Critic Agent causes the Critic Agent's function to fail, leading it to easily conform to the Env Agent's plot progression, losing control over the overall dialogue turns. At the same time, this result also confirms the effectiveness of RoleArena's innovative approach of introducing the Critic Agent. **2)** Using high-performance and similar-performance Agents can better ensure the stability of dialogue generation.
>
> | Language | Env Agent           | Critic Agent        | Success Rate |
> | -------- | ------------------- | ------------------- | ------------ |
> | **ZH**   | DeepSeek-V3.1       | DeepSeek-V3.1       | 95%          |
> |          | DeepSeek-V3.1       | Qwen2.5-7B-Instruct | **79%**      |
> |          | Qwen2.5-7B-Instruct | DeepSeek-V3.1       | 91%          |
> | **EN**   | DeepSeek-V3.1       | DeepSeek-V3.1       | 94%          |
> |          | DeepSeek-V3.1       | Qwen2.5-7B-Instruct | **75%**      |
> |          | Qwen2.5-7B-Instruct | DeepSeek-V3.1       | 90%          |
>
> ***Table 4**: Cross-Model Sensitivity Analysis for RoleArena*
>
> We sincerely thank you for pointing out this limitation. Addressing this issue helps improve the quality and completeness of our work, and we also hope that our results and explanations can resolve your concerns.
>
> ## Q'5: Addressing Concerns about Manual Annotation Efficiency
>
> Thank you for your valuable suggestions. To address your concerns, we provide explanations from both methodological and experimental perspectives.
>
> - Methodologically: Plot Progression in the RoleArena framework is carried out through the connection of discrete plot points. In principle, plot progression occurs when the Env Agent and Critic Agent reach consensus; at that moment, the Env Agent advances the plot based on two clearly defined discrete plot points. This explicit task allows the agents to naturally and autonomously generate smooth plot progression. Compared with fully manual annotation, our method greatly improves efficiency.
>
> - From the experimental perspective: In Table 2 of our rebuttal, we provide manual evaluation results of the dataset quality. We inspected 10% of the original dataset chosen through random sampling. The final results show an overall acceptance rate of 0.86 and an inter-annotator agreement of 0.84 measured by Krippendorff's Alpha.
>
> These results indicate that the dataset constructed based on RoleArena has high usability and reliability.
>
> ---
>
> **Finally**, we sincerely thank you for taking the time to review our manuscript and for providing high-quality comments and suggestions. We also appreciate your recognition of the contribution and innovation of our work. We hope that, based on your initial positive assessment, the discussion we have provided in response to your feedback can further strengthen your recommendation for accepting the manuscript. Thank you again!

---

### Official Review · Reviewer_MDvV · 2025-10-31

**Soundness:** 2
**Presentation:** 2
**Contribution:** 2
**Rating:** 2
**Confidence:** 3

**Summary:**

The paper presents 4 things: a role-playing environment (RoleArena), a dataset (PlotStream), models (PlotRole), and the evaluation system (RoleArena-Eval).

The environment contains: (i) an Environment agent that creates a topic, scene, and a discretized plotline of nodes, guides characters between turns, and decides when to progress to the next node; (ii) Character agents that produce thoughts, actions, and speech; and (iii) a Critic agent that overrides/accepts the Environment agent’s progression decisions to control the total number of turns given the remaining turns and nodes.

The Environment agent first creates a discretized plotline and guides character agents turn-by-turn with short narrations; after each character action, the environment agent decides whether to move to the next plot node. The critic agent vets these progression decisions using simple heuristics that consider remaining required turns r and remaining nodes, aiming to keep the total number of turns controllable and each node sufficiently developed

The dataset is based on interactions in this environment and has 39.3k characters and 170k dialogues (avg >50 turns) with markers for plot‑driving utterances (EN/ZH). The model used for the dataset creation is DeepSeek-V3.1.

The models are fine-tuned Qwen 2.5. Training simultaneously optimizes narration generation and character response generation.

The evaluation system has 5 Likert‑style LLM‑as‑a‑judge metrics.

Experiments across many open/closed LLMs and ablations (w/o env agent, story line, critic) suggest RoleArena improves generated dialogue quality and that PlotRole models outperform base Qwen models, with strong results particularly on Plot Progression. Limited human preference checks are provided.

**Strengths:**

Plot progression is a well-scoped and motivated problem. The focus on long multi-turn conversations is nice. A combination of discretized plot nodes with a critic agent enforcing turn-level pacing is a reasonable approach. The paper is easily understandable, though sometimes the text is too verbose.

**Weaknesses:**

1. The main weakness: it's not clear what problem this environment actually solves. Usually, in all role-playing applications, models interact with users. The primary complexity in building a role-playing benchmark lies in user simulation. This paper (and several other papers) just removes users from the picture. The remaining self-playing framework evaluates whether models are good at interacting with each other. But why should anyone care about that? One of the evaluation criteria mentions "tension to keep the user engaged", but there is no user as a part of the environment.

2. Evaluation is almost entirely LLM-as-judge; human evaluation is small-scale (50 role sets) and only alignment rates are reported without details on annotator agreement, protocol, or per-dimension reliability (Table 4). It is unclear how evaluation results translate to human satisfaction in real applications. The same with filtering, authors state that the dataset "has undergone manual verification to ensure data quality". Who were the annotators? What was the protocol?

3. The critic and plot-advance rules are heuristic and simplistic (Sec. 3.2.1–3.2.3): e.g., force progress if turns ≥6, or compare remaining turns with remaining nodes ×(3–5). No learning or adaptive mechanism; limited analysis of sensitivity to these thresholds."

4. Everything is generated by DeepSeek-V3.1. It is unclear whether models generalize beyond this synthetic distribution.

5. The procedure for discretizing plotlines, how key plot-driving sentences are marked, and how the narration versus character response targets are extracted need more algorithmic details or pseudo-code.


Minor points:
1. "Dialoge" in Table 1 and prompts, "RoArena-Eval" in line 145.

2. In Figures 2 and 3, there is no point in smoothing the loss curves.

**Questions:**

Suggestions:

1. Expand the human evaluation: at least 3 expert annotators, 100+ dialogues per language, per-dimension scoring with inter-annotator agreement (Krippendorff’s alpha). Carefully log the whole protocol. Report correlation with LLM judge per dimension and significance tests.

2. Demonstrate generalization beyond PlotStream distributions: evaluate with out-of-domain characters/scenes not generated by DeepSeek and report whether gains persist.

**Details Of Ethics Concerns:**

The conditions of human annotators are unclear.

---

> ### Author Response · Authors · 2025-11-23
> **Response to Reviewer MDvV (1/4)**
>
> We sincerely thank you for your valuable suggestions. These suggestions are very helpful for improving the completeness of our work and the quality of the manuscript. We will provide a detailed response to the points you have raised.
>
> ## W'1: Major Issues Addressed and User Involvement in the Research
>
> ### RoleArena: Issues Addressed
>
> Thank you for pointing this out. As stated in the motivation section, RoleArena addresses two main problems:
>
> 1. It provides a multi-agent collaborative environment for generating multi-turn role-playing dialogues with autonomous plot progression, offering a useful innovation over previous filtering and manual selection methods , as reviewer *GrBA* has also noted.
> 2. By introducing a critic agent combined with dynamic environmental interaction, RoleArena enables controllable, long multi-turn dialogues, filling the gap in current role-playing datasets that lack sufficient dialogue turns.
>
> Table 1 demonstrates that our PlotStream dataset achieves longer dialogue turns and a larger-scale plot-advancement corpus compared to existing datasets. Tables 2 and 6 show the performance improvements of PlotRole-7B and PlotRole-72B, validating RoleArena's effectiveness in constructing multi-turn role-playing data and enhancing plot-advancement capabilities.
>
> ### User Participation in RoleArena
>
> We acknowledge the key limitation of lacking direct user simulation in our proposed RoleArena and agree that simulating users is challenging. However, RoleArena does not ignore users; it addresses role-playing data scarcity through indirect bridging. Unlike existing benchmarks, which lack plot progression statements, rely on manual screening, and incur high costs with limited scale, RoleArena builds the PlotStream dataset via an innovative multi-agent collaborative framework. This enhances role-playing agents' plot progression, translating to greater user immersion by proactively simulating intentions rather than passive responses.
>
> To address your concerns, we used GPT-4o to simulate users in satisfied, dissatisfied, and preference modes across science fiction and horror styles. We ran experiments with untuned Qwen2-72B-Instruct and fine-tuned PlotRole-72B, recording dialogue turns between the simulated user and agent until reaching 5 turns or user termination.
>
> Table 1 reports results from 10 simulated runs over 100 scenes, showing that RoleArena's dataset improves plot progression, with average dialogue turns increasing by up to **62%**. We believe this will enhance user immersion in real-world scenarios.
>
> | Style  | Qwen2-72B-Instruct | PlotRole-72B | Δ Turns | Δ  Rate |
> | :----: | :----------------: | :----------: | :-----: | :-----: |
> | Sci-Fi |        2.9         |     4.7      |   1.8   | **62%** |
> | Horror |        3.1         |     4.3      |   1.2   | **39%** |
>
> ***Table 1**: Comparison of Dialogue Turns in User Simulation Scenarios Between Unfine-tuned Qwen2-72B-Instruct and PlotRole-72B*
>
> In addition, similar to IBSEN proposed by [1] Han et al. (2024), RoleArena is a scalable multi-agent collaborative environment; during the simulated dialogue process, any role agent can be replaced by the user for input, and at that time, the prompts given by the environment agent will serve as prompts for the user input.
>
> *[1]Han S, Chen L, Lin L M, et al. IBSEN: Director-Actor Agent Collaboration for Controllable and Interactive Drama Script Generation[C]//Proceedings of the 62nd Annual Meeting of the Association for Computational Linguistics (Volume 1: Long Papers). 2024: 1607-1619.*

---

> > ### Comment · Reviewer_MDvV · 2025-11-26
> >
> > The GPT-4o user simulation does not directly address my initial concern, and the performance of GPT-4o as a user simulator is not validated. However, the result is still nice and brings at least some connection to possible real-world applications.

---

> ### Author Response · Authors · 2025-11-23
> **Response to Reviewer MDvV (2/4)**
>
> ## W'2 & Q'1: Scaling of Manual Test Set Evaluation and Data Quality Assurance
>
> ### Expanded Evaluation Scale and Additional Results
>
> Thank you for your improvement suggestion regarding the scope of human evaluation.
>
> In response to your suggestion and to better demonstrate the evaluation results, we employed five expert annotators, each with two to three years of experience in role-playing. Scoring was based on the five dimensions proposed by the RoleArena-EVAL framework: DF: Dialogue Fluency, CF: Character Fidelity, EE: Emotional Expression, SQ: Story Quality, and PP: Plot Progression. Evaluation details for each dimension are provided in Section 5 of the paper. For consistency and rationality, we constructed 125 groups of dialogues for both Chinese and English scenarios using DeepSeek-V3.1, as mentioned in the paper. We selected K=15 to ensure the evaluation window length for each group falls within a reasonable range.
>
> In addition to GPT-4o, we selected **Claude Sonnet 4.5**, released by Google on September 29, 2025, as an additional large model critic to expand the human evaluation scope. **Table 2** reports the latest human evaluation results. The expert annotators show some deviation in preferences from the large model critic in the DF and EE dimensions, as annotators noted that some dialogues are slightly lengthy compared to normal human conversations, while the large model critic tends toward positive conclusions in emotional analysis.
>
> **Overall, the human evaluation results show strong correlation with the large model critic's evaluation results, and the human evaluation results exhibit high consistency**. The results confirm the rationality of the RoleArena-EVAL evaluation benchmark and the feasibility of using large models as evaluation experts.
>
> | Language | Dim      | Pearson (r) GPT-4o | p(GPT-4o) | Pearson (r) Claude-Sonnet-4.5 | p(Claude-Sonnet-4.5) | Krippendorff's Alpha |
> | -------- | -------- | ------------------ | --------- | ----------------------------- | -------------------- | -------------------- |
> | **EN**   | DF       | 0.812              | <0.01     | 0.785                         | <0.01                | 0.84, p<0.01         |
> |          | CF       | 0.909              | <0.01     | 0.927                         | <0.01                | 0.91, p<0.01         |
> |          | EE       | 0.804              | <0.01     | 0.833                         | <0.01                | 0.79, p<0.01         |
> |          | SQ       | 0.885              | <0.01     | 0.889                         | <0.01                | 0.88, p<0.01         |
> |          | PP       | 0.925              | <0.01     | 0.915                         | <0.01                | 0.92, p<0.01         |
> |          | Avg.(EN) | **0.867**          | -         | **0.869**                     | -                    | **0.87**             |
> | **ZH**   | DF       | 0.820              | <0.01     | 0.836                         | <0.01                | 0.82, p<0.01         |
> |          | CF       | 0.914              | <0.01     | 0.91                          | <0.01                | 0.89, p<0.01         |
> |          | EE       | 0.838              | <0.01     | 0.805                         | <0.01                | 0.78, p<0.01         |
> |          | SQ       | 0.905              | <0.01     | 0.906                         | <0.01                | 0.86, p<0.01         |
> |          | PP       | 0.919              | <0.01     | 0.925                         | <0.01                | 0.92, p<0.01         |
> |          | Avg.(ZH) | **0.879**          | -         | **0.876**                     | -                    | **0.85**             |
>
> ***Table 2**. Correlation and Agreement of Evaluation Results*
>
> ### Dataset Quality Control
>
> Thank you for highlighting the shortcomings in our PlotStream data quality control and acknowledge this issue. The quality control strategy is detailed in our response to reviewer *1Bbq*. **Table 3** reports the observed agreement rate Po, acceptance rate, expected agreement rate Pe, and Krippendorff's Alpha for inter-annotator agreement. The 86% acceptance rate and 0.84 Alpha confirm the dataset's reliability. Among rejected samples, 50% reached the long dialogue limit without ending; we removed these from the final PlotStream dataset to ensure quality.
>
> |  language   |    Po    | Accept Rate |    Pe    | Krippendorff's Alpha |  p-value   |
> | :---------: | :------: | :---------: | :------: | :------------------: | :--------: |
> |     zh      |   0.96   |    0.88     |   0.79   |         0.86         |   ＜0.01   |
> |     en      |   0.95   |    0.83     |   0.71   |         0.83         |   ＜0.01   |
> | **overall** | **0.96** |  **0.86**   | **0.75** |       **0.84**       | **＜0.01** |
>
> ***Table 3**: Manual Inspection Metrics for PlotStream*

---

> ### Author Response · Authors · 2025-11-23
> **Response to Reviewer MDvV (3/4)**
>
> ## W'3: Sensitivity Analysis of the Plot Progression Threshold
>
> Thank you for pointing this out; we acknowledge that the critic and plot progression rules are based on heuristics. However, this does not mean we overlooked the issue; specifically, we simplified the average dialogue turns required per plot point and achieved controllable dialogue generation through explicit heuristic rules.
>
> To better address your concerns about the necessity of forcing plot progression, we conducted a sensitivity analysis on the thresholds. In **Table 4**, we report the average results from sampling 50 groups of dialogues 10 times in the English scenario. The results confirm that RoleArena maintains an accuracy rate above 90% for data generation under different plot progression rule requirements, verifying the stability of RoleArena in generating dialogues. Without adding the forced progression rules, the success rate **decreased by up to 21%**, which demonstrates the necessity of the forced plot progression rules.
>
> It should be noted that successful dialogue generation refers to the entire dialogue successfully completing the storyline and meeting the minimum required dialogue turns, and Avg Turns represents the average number of dialogue turns per plot point.
>
> | Setting            | Max Turns | Turns Range | Success rate | Avg Turns |
> | ------------------ | --------- | ----------- | ------------ | --------- |
> | **base**           | 6         | (3,5)       | 93%          | 4.3       |
> |                    | 5         | (2,4)       | 92%          | 3.5       |
> |                    | 7         | (4,6)       | 93%          | 5.1       |
> | **w/o Constraint** | 6         | (3,5)       | 72%          | 4.1       |
> |                    | 5         | (2,4)       | 78%          | 3.6       |
> |                    | 7         | (4,6)       | 75%          | 5.5       |
>
> ***Table 4**: Sensitivity Analysis of the Threshold for RoleArena Dialogue*
>
> Heuristic learning for the average dialogue turns across different plot points is not our core contribution; this paper focuses more on the generation of long-turn multi-turn dialogues with plot progression. In the future, adaptive learning for dialogue turn thresholds can be achieved through lightweight prediction models, and we also warmly welcome future research work conducted using PlotStream.
>
> ## W'4 & Q'2: Demonstrating the Generalization Capability of RoleArena
>
> Thank you for your valuable suggestions. To address your questions, we used **Gemini-2.5** and **Claude-Sonnet-4.5** to generate 100 groups of out-of-domain roles, scenes, and plotlines separately in Chinese and English scenarios, followed by using DeepSeek-V3.1 as mentioned in the paper to generate role-playing multi-turn dialogues under the RoleArena framework.
>
> In **Table 5**, we report the evaluation results based on GPT-4o. From these results, it can be seen that RoleArena has good generalization ability. In addition, the additionally reported dialogue generation success rates under different settings are consistently greater than 90%, which confirms that RoleArena has strong robustness in dialogue generation while maintaining generalization ability.
>
> | Language | OOD LLM               | LLM               | DF    | CF    | EE    | SQ    | PP    | Average   | Success Rate |
> | -------- | --------------------- | ----------------- | ----- | ----- | ----- | ----- | ----- | --------- | ------------ |
> | **ZH**   | -                     | **DeepSeek-V3.1** | 4.220 | 4.158 | 4.028 | 4.672 | 4.566 | **4.329** | 95%          |
> |          | **Gemini-2.5**        |                   | 4.351 | 4.026 | 4.210 | 4.585 | 4.627 | **4.360** | 94%          |
> |          | **Claude-Sonnet-4.5** |                   | 4.229 | 4.165 | 4.108 | 4.694 | 4.487 | **4.337** | 94%          |
> | **EN**   | -                     | **DeepSeek-V3.1** | 4.406 | 4.564 | 4.550 | 4.912 | 4.550 | **4.596** | 93%          |
> |          | **Gemini-2.5**        |                   | 4.159 | 4.338 | 4.439 | 4.754 | 4.564 | **4.451** | 93%          |
> |          | **Claude-Sonnet-4.5** |                   | 4.025 | 4.135 | 4.208 | 4.652 | 4.690 | **4.342** | 94%          |
>
> ***Table 5**. Generalization Results of RoleArena*

---

> ### Author Response · Authors · 2025-11-23
> **Response to Reviewer MDvV (4/4)**
>
> ## W'5: Algorithm Pseudocode for Discretizing and Labeling Key Plot Points
>
> Thank you very much for your valuable suggestions; in the latest **revised version** of the paper's **Appendix E**, we added pseudocode for the overall operation of the RoleArena framework, and in the pseudocode, we highlighted the plotline discretization (**line 4**) and key plot progression (**line 29**).
>
> The pseudocode does not directly provide the step for annotating the character's response during key plot progression; this is because when constructing the training data for PlotRole, we can directly select the next character's utterance after the key plot annotation, which serves as the character's response during plot turning progression, so there is no need to specially annotate this statement in the RoleArena algorithm process.
>
> ### Minor Points
>
> Thank you very much for pointing out some minor issues in the paper; in the revised version, we corrected the accurate terminology for “Dialoge” and “RoleArena-Eval”. At the same time, in Appendix C, we removed the display of the smoothed loss curve, correspondingly revised the related descriptions, and marked them prominently with red font. Thank you again for your feedback.
>
> ---
>
> **Finally**, we sincerely thank you for the valuable time spent reviewing the manuscript and for the important suggestions provided. Thank you for affirming our research motivations and ideas. We hope that our responses can further improve the quality of the manuscript; we are very willing to communicate further with you, and if you have any other questions, please feel free to contact us at any time.

---

> > ### Comment · Reviewer_MDvV · 2025-11-26
> >
> > Thank you for the comments and the new revision! Overall, they address most of my concerns, so I'm increasing my score from 2 to 6.

---

> > > ### Author Response · Authors · 2025-11-26
> > >
> > > We sincerely thank you for the positive evaluation and score improvement. If you have any other questions, please feel free to contact us.
> > > Once again, we sincerely thank you for your valuable feedback, which is crucial for improving the quality of our paper.

---

### Official Review · Reviewer_1Bbq · 2025-11-01

**Soundness:** 2
**Presentation:** 2
**Contribution:** 3
**Rating:** 4
**Confidence:** 5

**Summary:**

This paper proposes a novel multi-agent role-playing framework named RoleArena, designed to generate long multi-turn dialogues with autonomous plot progression.

Traditional role-playing language agent (RPLA) datasets are insufficient in dialogue depth and plot progression. To address this, RoleArena introduces three types of agents: character agents, environment agents, and critic agents. The environment agent is responsible for generating discretized plot lines and dynamically judging when to advance the plot. Meanwhile, the critic agent provides a secondary evaluation of the environment agent's decisions by assessing the remaining dialogue turns and plot complexity, ensuring that the dialogue is sufficiently developed and the total turns are controllable. The critic agent is one of the key innovations in the paper.

Based on RoleArena, the authors constructed a large-scale dataset, PlotStream, and trained the PlotRole series models. Furthermore, the paper proposes a five-dimension evaluation benchmark, RoleArena-Eval, which specifically introduces a quantitative assessment of plot progression capability.

**Strengths:**

The paper identifies a key shortcoming in existing RPLA datasets and models: the lack of active and logical plot progression in long multi-turn dialogues. This work presents not just an environment, but a complete package: a large-scale dataset (PlotStream), stronger models trained on it (PlotRole), and a targeted evaluation benchmark (RoleArena-Eval). The paper's ablation studies effectively demonstrate the necessity of each component.

**Weaknesses:**

1. **Insufficient evidence of dataset quality**: The paper provides almost no description of how the PlotStream dataset was manually checked and verified.
2. **Concerns about data diversity**: We note that the paper's method relies on the Environment agent pre-generating "discretized plot lines," which are then fleshed out to enrich the plot. Does this lead to similar or repetitive plots under different settings? This concern is twofold: (1) Are the storylines in the constructed dataset repetitive or strongly biased towards a certain style? We note the authors emphasize ensuring data diversity for character profiles, but there seems to be a gap between this and storyline diversity. (2) Can the RoleArena-Eval benchmark detect plot repetitiveness? The evaluation system lacks a measure for Inter-story Diversity.
3. **Formatting and citation issues**: There are several oversights, such as the missing citation on Line 435 and issues with the formatting of the References(e.g., the same article appears twice in the reference list).

**Questions:**

1. Could you provide more details on the human verification of PlotStream's quality? For example, the verification criteria, acceptance rate, and inter-annotator agreement (IAA)?
2. Given that the system uses pre-generated "discretized plot lines," how did you ensure the plots in the PlotStream dataset are diverse and not repetitive? Did you analyze the distribution or novelty of the plotlines in the dataset?

---

> ### Author Response · Authors · 2025-11-23
> **Response to Reviewer 1Bbq (1/3)**
>
> We sincerely thank you for your valuable comments, which are very important for improving the overall quality of our paper. Next, we provide detailed responses to the concerns you raised.
>
> ## W'1 & Q'1: PlotStream Dataset Validation Description
>
> Thank you for highlighting the shortcomings in our manual validation of the PlotStream dataset; we acknowledge this issue. Below, we outline the process.
>
> We hired 5 professional annotators with 2–3 years of role-playing experience. We randomly sampled 500 dialogue sets from PlotStream and assigned each annotator 135 Chinese and 135 English sets, covering ~10% of dialogue turns overall, with 2.6 annotators per set for overlap review. Annotators scored across RoleArena-Eval's five dimensions; predictive experiments set a 3.0 rejection threshold (or below 2.0 per dimension).
>
> In Table 1, we report observed agreement (Po), acceptance rate, and expected agreement (Pe), using Krippendorff's Alpha (0.84) for inter-annotator agreement. The 86% acceptance rate confirms PlotStream's quality. Of rejected samples, 50% failed to end after max turns; we removed these from the final dataset. In addition, using the same rejection threshold as manual evaluation, when finalizing the PlotStream dataset, we used GPT-4o combined with RoleArena-Eval to remove data with composite scores below 3 in the dataset.
>
> |  language   |    Po    | Accept Rate |    Pe    | Krippendorff's Alpha |     p      |
> | :---------: | :------: | :---------: | :------: | :------------------: | :--------: |
> |     ZH      |   0.96   |    0.88     |   0.79   |         0.86         |   ＜0.01   |
> |     EN      |   0.95   |    0.83     |   0.71   |         0.83         |   ＜0.01   |
> | **overall** | **0.96** |  **0.86**   | **0.75** |       **0.84**       | **＜0.01** |
>
> ***Table 1**: Manual Inspection Metrics for PlotStream*

---

> ### Author Response · Authors · 2025-11-23
> **Response to Reviewer 1Bbq (2/3)**
>
> ## W'2 & Q'2: Diversity Check for Discretized Plot Lines(Part.1)
>
> ### Plot Line Diversity Analysis
>
> To verify story line diversity across styles, we computed similarities using two top MTEB multilingual Leaderboard embedding models: **Qwen3-Embedding-8B** and **KaLM-Embedding-Gemma3-12B-2511**.
>
> For efficient comprehensive reporting, we ran 10 samplings of 100 dialogue sets each and averaged results in **Tables 2** and **3**. Key findings: PlotStream plot lines show no high similarity, with a maximum average of 0.4422, confirming their diversity.
>
> The diversity of PlotStream story lines stems from these measures:
>
> 1. For PersonHub-based role selection, we chose orthogonal roles via embeddings and defined dual personalities plus five major traits per role to enrich content.
> 2. Drawing from literature and film typologies, we divided roles into 10 styles to control diverse story generation; per style, we sampled groups and generated discrete lines, with details in **Appendix D.1**.
> 3. Varied hyperparameters: In RoleArena-based generation, we dynamically selected random seeds and temperature coefficients to increase randomness.
>
> | Model              | Style(EN)       | SIM(EN)    | Style(ZH)       | SIM(ZH)    |
> | ------------------ | --------------- | ---------- | --------------- | ---------- |
> | Qwen3-Embedding-8B | Sci-Fi          | 0.4105     | Sci-Fi          | 0.4293     |
> |                    | Fantasy         | 0.4110     | Fantasy         | 0.4198     |
> |                    | Workplace Drama | 0.3812     | Workplace Drama | 0.3506     |
> |                    | Historical      | 0.3480     | Historical      | 0.3442     |
> |                    | Horror          | 0.3995     | Horror          | 0.4251     |
> |                    | Romance         | 0.3544     | Romance         | 0.3595     |
> |                    | Mystery         | 0.4155     | Mystery         | 0.4315     |
> |                    | Adventure       | 0.4082     | Adventure       | 0.4307     |
> |                    | Urban Fiction   | 0.3146     | Urban Fiction   | 0.3407     |
> |                    | Superpower      | 0.4238     | Superpower      | 0.4536     |
> | **Average**        | **-**           | **0.3867** | -               | **0.3985** |
>
> ***Table 2**: Plot Line Similarity Computation Results Based on Qwen3-Embedding-8B*

---

> ### Author Response · Authors · 2025-11-23
> **Response to Reviewer 1Bbq (3/3)**
>
> ## W'2 & Q'2: Diversity Check for Discretized Plot Lines(Part.2)
>
> | Model                          | Style(EN)        | SIM(EN)     | Style(ZH)        | SIM(ZH)     |
> | ------------------------------ | --------------- | ---------- | --------------- | ---------- |
> | KaLM-Embedding-Gemma3-12B-2511 | Sci-Fi          | 0.4738     | Sci-Fi          | 0.5049     |
> |                                | Fantasy         | 0.4681     | Fantasy         | 0.4877     |
> |                                | Workplace Drama | 0.4052     | Workplace Drama | 0.3981     |
> |                                | Historical      | 0.4116     | Historical      | 0.3960     |
> |                                | Horror          | 0.4536     | Horror          | 0.4440     |
> |                                | Romance         | 0.4049     | Romance         | 0.4139     |
> |                                | Mystery         | 0.4439     | Mystery         | 0.4458     |
> |                                | Adventure       | 0.4443     | Adventure       | 0.4790     |
> |                                | Urban Fiction   | 0.3789     | Urban Fiction   | 0.3874     |
> |                                | Superpower      | 0.4515     | Superpower      | 0.4654     |
> | **Average**                    | **-**           | **0.4336** | -               | **0.4422** |
>
> ***Table 3**: Plot Line Similarity Computation Results Based on KaLM-Embedding-Gemma3-12B-2511*
>
> ### Insufficient Evaluation of Plot Repetition in RoleArena-Eval
>
> Thank you for pointing this out. We acknowledge that the current RoleArena-Eval benchmark has limitations in checking repetition between plot lines. RoleArena-Eval mainly evaluates five dimensions—dialogue fluency, character fidelity, emotional expression, story quality, and plot progression—using LLMs as judges. Attempting to include plot repetition detection in the story quality dimension raises these issues:
>
> - **Greatly Increases Evaluation Difficulty**. RoleArena-Eval assesses quality via analysis of characters, themes, environments, and global dialogues, covering key focus areas in role-playing. Introducing other story lines requires LLMs to compute similarities and judge them, needing integration of numerous comparison cases.
> - **Difficulty in Achieving Complete Plot-Line Coverage**. A single evaluation must incorporate plot lines from all other samples; this becomes infeasible for large-scale datasets.
>
> Thus, we recommend plot-line repetition detection as a separate metric in the dataset quality process. Using advanced embedding models, we compute cosine similarity between plot line embeddings to measure repetition levels. Thank you again for your comment.
>
> ## W'3: Citation errors and duplication of literature
>
> Thank you for pointing out the format error; in the latest **revised** **revision** version, we have added the citation for **Table 4**, which can be seen specifically on line 448-449, and we have highlighted the corresponding text in blue. For the error of literature duplication, we have revised it and correctly cited the reference [1] Zhou et al. (2024). Thank you again for your detailed review; we have checked the other references throughout the full text.
>
> *[1] Zhou J, Chen Z, Wan D, et al. CharacterGLM: Customizing Social Characters with Large Language Models[C]//Proceedings of the 2024 Conference on Empirical Methods in Natural Language Processing: Industry Track. 2024: 1457-1476.*
>
> ---
>
> **Finally**, we sincerely thank you for taking valuable time to review our manuscript and providing highly important, quality feedback. We also appreciate your recognition of the innovations and contributions in our work. We are very pleased to have the opportunity to continue discussions with you to improve the overall quality of our paper, and if you have any further questions, we are very willing to maintain open communication.

---

### Author Response · Authors · 2025-11-29
**Rebuttal Summary**

## Discussion Summary

We sincerely thank all reviewers for their constructive comments and valuable time. We released the rebuttal content on **November 23(EST)**. As of now, we have received responses from (2/4) reviewers：

- Reviewer *MDvV* stated that we addressed most of his concerns and **raised the** **score from 2 to 6** on **November 26(EST)**.  At the same time, our overall scores became **6,6,4,4**. We thank the reviewer for the understanding and the score increase.
- Reviewer *8Dqz* still has questions about innovation and dataset quality and maintains a score of 4 on November 25(EST). We have not yet received a new response to our further explanations. We thank the reviewer for the opportunity to communicate and hope that our explanations can resolve his concerns.

## Concerns Resolution Summary

We have provided complete responses to the reviewers' concerns. Here is a summary:

1. **Data quality inspection** (for Reviewers *1Bbq*, *MDvV*, *GrBA*, *8Dqz*). We report the experimental results from manual dataset inspection and quality control.
2. **Human evaluation scale and multi-model assessment** (for Reviewers *MDvV*, *GrBA*, *8Dqz*). We report experiments using multiple models (GPT-4o and Claude-Sonnet-4.5) for evaluation and have expanded the size of the human evaluation test set.
3. **Plotline diversity** (for Reviewer *1Bbq*). We report the plotline diversity analysis results and have included it as a new evaluation metric.
4. **User participation** (for Reviewer *MDvV*). We report the results from simulation experiments on user participation within RoleArena.
5. **Plot progression threshold sensitivity** (for Reviewers *MDvV*, *GrBA*). We report the experimental results from the sensitivity analysis on the plot progression threshold.
6. **Cross-model sensitivity** (for Reviewer *GrBA*). We report the cross-model sensitivity analysis results for RoleArena.
7. **RoleArena generalization ability** (for Reviewers *MDvV*, *8Dqz*). We report the experimental results on the generalization capability of RoleArena.
8. **Evaluation scope** (for Reviewer *8Dqz*). We have expanded the model evaluation scope and report the corresponding test results.
9. **Dataset statistics** (for Reviewer *8Dqz*). We report the statistical results for the PlotStream dataset, which are updated in **Appendix D.2** of the latest revision.
10. **Base model evaluation** (for Reviewer *8Dqz*). We report the evaluation results for Qwen2.5-7B and Qwen2.5-72B.
11. We have: ***1)*** added the RoleArena algorithm pseudocode in **Appendix E** (for Reviewer *MDvV*), ***2)*** explained the model consensus mechanism, the rationale for comparing RoleArena with CharacterBox, the specific training objectives, and the advantages of RoleArena (for Reviewer *GrBA*), ***3)*** elaborated on the innovative aspects of RoleArena (for Reviewer *8Dqz*).
12. **Citation formatting and text corrections** (for Reviewers *1Bbq*, *MDvV*, *GrBA*, *8Dqz*). We have corrected citations, table references, and some terminology errors in the latest revision.

## Highlights Summary

We are encouraged that the reviewers expressed recognition of our work in several aspects. Here is a summary:

- **Novel problem**: Reviewers *1Bbq*, *GrBA*, and *8Dqz* stated that we identified a key shortcoming in the current RPLA field: the lack of research on plot progression in long multi-turn dialogues.To address this challenge, we proposed a complete solution.
- **Novel approach**: Reviewers *1Bbq* and *GrBA* acknowledged the novelty of our approach. We note that RoleArena introduces discrete plot lines and turn-level dynamic plot judgments, and introduces a critic agent combined with discrete plot lines to achieve turn-controllable long-turn autonomous plot progression in role-playing multi-turn dialogues.
- **Important contributions**: Reviewer *GrBA* stated that the PlotStream and RoleArena-Eval proposed in this paper are valuable resources for the community.
- **Sufficient motivation**: Reviewers *MDvV* and *8Dqz* noted that our work has clear motivation and well-defined problems.
- **Effective method**: Reviewer *MDvV* indicated that the RoleArena solution is reasonable , and Reviewer *GrBA* noted that our comprehensive experiments confirm the method's effectiveness.
- **The paper is easily understandable**: Reviewer *MDvV* stated that our paper is easy to understand.

---

Finally, we sincerely thank all the reviewers, AC, SAC, and PC for their valuable time spent on this paper.

We also extend our sincere gratitude to the reviewers for their valuable comments, which have greatly helped us improve the quality of the paper.

Best regards,



The authors of Paper 7108

---

### Meta-Review · Area_Chair_jVJA · 2026-01-07

**Summary:**

This work got 2 marginally below, 1 reject, 1 marginally above ratings. The author rebuttals are extensive, with diverse new experimental results and analyses. The fix list the authors provided is substantially long. In the mid of processes, some reviewers raised their ratings, thus the work got 6, 6, 4, 4, a borderline case. As a piece of work proposing a pipeline to dialogues generation, e.g. the data quality inspection, human evaluation, diversity, plots, cross-modal sensitivity, generalisation issues, evaluation scopes and data stats, diverse base models, are all crucial and should have been in place of the original manuscript than being added during the rebuttals. ACs find it hard to judge in its present form, and recommend the authors reflect all the concerns to the original manuscript and resubmit to other similar venues for re-evaluations.

**Reviewer Concerns:**

The same as above.

**Reviewer Scores:**

This work got 2 marginally below, 1 reject, 1 marginally above ratings. The author rebuttals are extensive.

---

### Decision · Program_Chairs · 2026-01-26

Reject